# PromptFE: Automated Feature Engineering by Prompting

## Abstract

Automated feature engineering (AutoFE) liberates data scientists from the burden of manual feature construction. The semantic information of datasets contains rich context information for feature engineering but has been underutilized in many existing AutoFE works. We present PromptFE, a novel AutoFE framework that leverages large language models (LLMs) to automatically construct features in a compact string format and generate semantic explanations based on dataset descriptions. By learning the performance of constructed features in context, the LLM iteratively improves feature construction. We demonstrate through experiments on real-world datasets the superior performance of PromptFE over state-of-the-art AutoFE methods. We verify the impact of dataset semantic information and provide comprehensive study on the LLM-based feature construction process.

## 1 Introduction

Tabular data, a form of structured data comprising instances and attributes, have extensive use across a broad range of domains including credit assessment, market prediction, and quality control. Traditional machine learning models, especially tree-based models (Breiman, 2001; Ke et al., 2017), have strong performance on tabular datasets of small and medium sizes (Grinsztajn et al., 2022) and good interpretability. Feature engineering refers to the construction of features to enhance the performance of downstream models, which is crucial for traditional ML models as new features extract useful information for target prediction by capturing complex non-linear relationships. Feature engineering by hand demands domain expertise to relieve significant human labor.

Automated feature engineering (AutoFE) employs meta algorithms and models to automate feature engineering process for performance comparable to domain experts. Prior approaches like (Zhu et al., 2022a;b; Zhang et al., 2023) construct and evaluate enormous features in a trial-and-error manner. While some methods learn to optimize the utility of features during the FE process, they do not utilize domain knowledge to guide feature search. The need to search features from scratch for new datasets and downstream models hampers their efficacy and efficiency. Furthermore, these methods cannot offer explanation of the engineered features, undermining the interpretability.

The text descriptions of tabular datasets provide rich context for feature engineering. Domain experts consult attribute descriptions to select relevant feature attributes and compute new features useful for target prediction. For example, the *square footage* of a house times the *average housing price per square foot* in the neighborhood may be a good predictor of the *market value* of the house. Pretrained on large volumes of data, large language models (LLMs) (Radford et al., 2019; Brown et al., 2020; OpenAI, 2023; Touvron et al., 2023a;b) handle general language processing tasks and encapsulate extensive domain knowledge. Under proper instructions, an LLM can process dataset semantic information and utilize its knowledge to au-

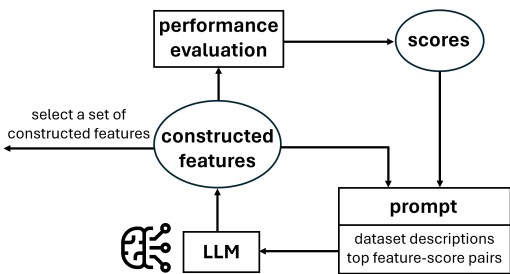

**Figure 1:** Overview of PromptFE: (1) instructing the LLM to construct new features by providing dataset descriptions and example features; (2) evaluating the constructed features; (3) updating the prompt with top-performing features and scores; and (4) selecting a set of constructed features to add to the dataset.

tomatically construct features in a manner similar to domain experts. The work by Hollmann et al. (2023) demonstrates the potential of such research direction but is not sufficiently effective in feature search. Similarly, the work by Nam et al. (2024) suffers from large search space. The works by Han et al. (2024) and Zhang et al. (2024b) do not involve feature learning and improvement.

We present Auto**FE** by **Prompt**ing (PromptFE), a novel AutoFE framework that leverages LLMs for effective, efficient, and interpretable feature engineering, as illustrated in Figure 1. With dataset descriptions and example features in canonical Reverse Polish Notation (cRPN), we prompt the LLM to construct new features. After evaluation, we update the prompt with top-performing features with the evaluation scores and instruct the LLM to construct further features. Iteratively, the LLM explores the feature space and improves solutions by learning good examples in context. The dataset semantic information not only guides feature search, but helps the LLM understand the patterns in example features. Applying domain knowledge, the LLM generates semantically meaningful features and explains their usefulness. Experiments on real-world datasets demonstrate that PromptFE yields over 5% mean performance gain for three downstream models and significantly outperforms state-of-the-art baselines. Furthermore, we show in ablation study the effects of dataset semantic context and proposed feature canonicalization scheme. We also comprehensively study the behavior of the LLM-based feature construction process.

Our main contributions are: (1) We introduce a novel LLM-based AutoFE framework utilizing dataset semantic information for automated feature construction, which is the first method capable of generating features in the RPN format while providing semantic explanations. (2) We benchmark the performance of our approach against state-of-the-art baselines using both GPT-3.5 and GPT-4. (3) We investigate the impact of semantic context and study the behavior of the LLM-based feature construction process, providing a comprehensive view of our approach.

## 2 RELATED WORK

**Large Language Models.** LLMs are large-scale general-purpose neural networks pretrained on vast corpora of text data, typically built with transformer-based architectures (Vaswani et al., 2017). Generative LLMs, such as the GPT family (Radford et al., 2019; Brown et al., 2020; OpenAI, 2023) and the LLaMA family (Touvron et al., 2023a;b), are pretrained to successively generate the next token given the text input and can be finetuned using reinforcement learning from human feedback (Ziegler et al., 2019; Ouyang et al., 2022). By this means, they acquire the syntactic and semantic knowledge of natural languages and achieve state-of-the-art performance on various tasks including text generation, summarization, and question answering. Prompting techniques (Liu et al., 2023) have been developed to adapt LLMs to downstream tasks without modifying model weights. Few-shot learning (Brown et al., 2020) includes examples in the prompt for the language model to learn in context. Leveraging such capability, an LLM may function as a problem solver (Yang et al., 2024) that iteratively improves candidate solutions according to the task description and performance feedback. Chain-of-thought (Wei et al., 2022; Kojima et al., 2022) strengthens reasoning performance of LLMs through the elicitation of intermediate reasoning steps.

**Automated Feature Engineering.** AutoFE complements the input dataset with engineered features to enhance the performance of downstream models. Traditional AutoFE approaches include expansion-reduction (Kanter & Veeramachaneni, 2015; Horn et al., 2020; Zhang et al., 2023), evolutionary algorithms (Smith & Bull, 2005; Zhu et al., 2022a), and reinforcement learning (Khurana et al., 2018; Li et al., 2023; Wang et al., 2023). DIFER (Zhu et al., 2022b) utilizes encoder-decoder neural networks to learn the utility of features and optimize features in the embedding space. OpenFE (Zhang et al., 2023) develops a feature boost algorithm to speedup feature evaluation. Nonetheless, these traditional approaches do not incorporate the semantic information of datasets, which hampers the efficacy and interpretability of engineered features.

**AutoFE with Domain Knowledge.** The benefits of incorporating domain knowledge in AutoFE include: (1) improving the effectiveness; and (2) reducing the cost of feature search, especially the feature evaluation overhead. One direction in prior works is to learn transferrable knowledge. LFE (Nargesian et al., 2017) represents features with quantile sketches transferable across datasets and inputs them to a feature transformation recommendation model. FETCH (Li et al., 2023) is an RL-based AutoFE framework that takes tabular data as the state and is generalizable to new data. E-

AFE (Wang et al., 2023) pretrains a feature evaluator to help efficiently train the RL-based AutoFE model. The other direction is to leverage the semantic information of datasets. KAFE (Galhotra et al., 2019) employs knowledge graphs to identify semantically informative features relevant to the prediction task. CAAFE (Hollmann et al., 2023) manipulates datasets using the code generated from an LLM based on dataset descriptions. FeatLLM (Han et al., 2024) generates first-order rules for classification tasks. ELF-Gym (Zhang et al., 2024b) generates first feature descriptions and then feature code. Neither approach involves feature learning and improvement. OCTree (Nam et al., 2024) relies on external decision tree algorithms to represent features and suffers from large search space. Differently, we adopt a compact form of feature representation in cRPN with pre-defined transformation operators. Our approach reduces the search space and helps the LLM learn the patterns of useful features, leading to stronger and more robust performance.

## 3 NOTATIONS

We denote a tabular dataset as $D = \langle \mathbb{X}, \boldsymbol{y} \rangle$, where $\mathbb{X} = \{\boldsymbol{x}_1, \ldots, \boldsymbol{x}_d\}$ is the set of raw features with $\boldsymbol{x}_i \in \mathbb{R}^n$ for $i = 1, \ldots, d$ and $\boldsymbol{y} \in \mathbb{R}^n$ is the target. We construct a new feature $\tilde{\boldsymbol{x}} = t(\boldsymbol{x}_{j_1}, \ldots, \boldsymbol{x}_{j_o})$ by transforming existing features $\boldsymbol{x}_{j_1}, \ldots, \boldsymbol{x}_{j_o}$ via some operator $t \in \mathbb{R}^n \times \ldots \times \mathbb{R}^n \to \mathbb{R}^n$ of arity $o$. Given a set of transformation operators $\mathbb{T}$, we define the feature space $\mathbb{X}_{\mathbb{T}}$ recursively as: for any $\tilde{\boldsymbol{x}} \in \mathbb{X}_{\mathbb{T}}$, either $\tilde{\boldsymbol{x}} \in \mathbb{X}$; or $\exists t \in \mathbb{T}$, s.t., $\tilde{\boldsymbol{x}} = t(\tilde{\boldsymbol{x}}_{j_1}, \ldots, \tilde{\boldsymbol{x}}_{j_o})$, where $\tilde{\boldsymbol{x}}_{j_1}, \ldots, \tilde{\boldsymbol{x}}_{j_o} \in \mathbb{X}_{\mathbb{T}}$. To measure feature complexity, we compute the order of a feature $\tilde{\boldsymbol{x}} \in \mathbb{X}_{\mathbb{T}}$ as:

$$\alpha(\tilde{\boldsymbol{x}}) = \begin{cases} 0 & \text{if } \tilde{\boldsymbol{x}} \in \mathbb{X}, \\ 1 + \max_j \alpha(\tilde{\boldsymbol{x}}_j) & \text{if } \tilde{\boldsymbol{x}} = t(\tilde{\boldsymbol{x}}_{j_1}, \ldots, \tilde{\boldsymbol{x}}_{j_o}) \ \exists t \in \mathbb{T}. \end{cases} \quad (1)$$

The constrained feature space with the order upper bounded by $k$ is denoted as $\mathbb{X}_{\mathbb{T}}^{(k)} = \{\tilde{\boldsymbol{x}} \in \mathbb{X}_{\mathbb{T}} \mid \alpha(\tilde{\boldsymbol{x}}) \leq k\}$.

We denote the performance of a downstream machine learning model algorithm $M$ on the dataset as $\mathcal{E}_M(\mathbb{X}, \boldsymbol{y})$. The objective of AutoFE is to augment the dataset with a set of constructed features $\tilde{\mathbb{X}}^*$ to optimize the model performance, specifically:

$$\tilde{\mathbb{X}}^* = \arg\max_{\tilde{\mathbb{X}} \subset \mathbb{X}_{\mathbb{T}}} \mathcal{E}_M(\mathbb{X} \cup \tilde{\mathbb{X}}, \boldsymbol{y}). \quad (2)$$

## 4 METHODOLOGY

In this section, we present PromptFE, a novel AutoFE framework leveraging the power of LLMs, particularly, the GPT models (Radford et al., 2019; Brown et al., 2020; OpenAI, 2023). The high-level idea is to provide the LLM with descriptive information of the dataset in the prompt and guide it to search for effective features using examples.

We represent features in a compact form in our prompt. A feature $\tilde{\boldsymbol{x}} \in \mathbb{X}_{\mathbb{T}}$ is expressible as a tree, where the leaf nodes are raw features and the internal nodes are operators. However, the expression trees of features containing commutative operators (like addition and multiplication) are not unique since the child nodes of these operators are unordered. We introduce a canonicalization scheme: arranging operator nodes before

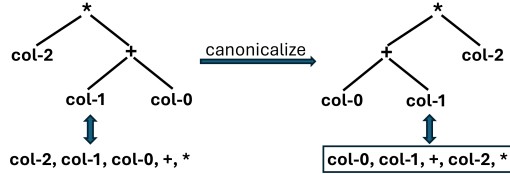

**Figure 2:** We obtain canonical RPN (cRPN) by re-ordering the nodes of a feature expression tree.

feature nodes for left skewness and lexicographically sorting the nodes within each group. We then serialize the canonical expression tree into the postorder depth-first traversal string, i.e., canonical reverse Polish notation (cRPN), ensuring the one-to-one mapping between features and string representations. We denote the feature corresponding to an RPN string $f$ as $\tilde{\boldsymbol{x}}_f$ and the set of features corresponding to a set of RPN strings $\mathbb{F}$ as $\tilde{\mathbb{X}}_{\mathbb{F}}$. We make further discussions in Appendix A.

Our prompt contains: (1) a meta description of the dataset; (2) an indexed list of the dataset attributes, with attribute types, value ranges, and descriptions; (3) lists of transformation operators with descriptions, grouped by the arity; (4) a ranked list of example features with performance evaluation scores; (5) feedback of previously constructed features; and (6) an output template of new

---

(1) Dataset description:
This dataset contains information on default payments, demographic factors, credit data, and history of payment of credit card clients ...
(2) Dataset contains the following columns:
col-0 (int) [10000, 800000]: LIMIT_BAL: Amount of given credit in NT dollars (includes individual and family/supplementary credit
col-1 (category) {1, 2}: SEX: Gender (1=male, 2=female) . . .
col-23 (category) {0, 1}: default.payment.next.month: Default payment (1=yes, 0=no)
(3) We have the following unary operators:
sqrt_abs: taking the square root of the absolute value . . .
We have the following binary operators:
+: summing two columns . . .
Feature strings are reverse Polish notation (RPN) expressions that operate on the columns of our dataset. Each feature string constructs an extra
column that is useful for the downstream model Random Forests to predict the target col-23. The model will be trained on the dataset with the
constructed columns and evaluated on a holdout set. The best columns will be selected.
(4) Below are feature strings arranged in ascending order based on their performance scores. Higher scores are better.
Feature
col-17,col-21,*,col-20,+,sqrt_abs
Score
0.0011 . . .
Feature
col-4,col-6,*,col-12,col-16,-,sqrt_abs,*
Score
0.0014
(5) Previous feature:
col-4,col-5,col-6,+,col-12,col-16,-,*
Error: invalid RPN expression
Give me a new feature string that is different from all strings above and has a higher score. Use no more than five operators. Make sure all
columns and operators exist and do not include the target column. Follow the syntax of RPN.
(6) Output format:
Feature
(Feature description)
Usefulness
(Explanation why this adds useful real world knowledge to predict the target col-23 according to dataset description)

**Figure 3:** Prompt template. Sections containing dataset information are marked in blue. The ranked list of feature examples and scores is marked in orange. The feedback message is marked in gray.

---

Feature
col-11,col-11,col-12,-,log,*
This feature calculates the log of the difference between the September bill statement (col-11) and the August bill statement (col-12), then
multiplies it by the September bill amount.
Usefulness
This feature captures the change in the bill amount from August to September in a logarithmic scale, which can effectively highlight significant
changes in spending patterns. Large fluctuations in credit card bills could be indicative of financial distress, which may impact the likelihood of
defaulting on payments, while the logarithmic transformation allows for handling potential skewness in the data distribution. By incorporating
this feature, the model can better understand how temporal changes in spending behavior relate to the probability of default, providing richer
contextual information beyond static features.

**Figure 4:** The LLM constructs a new feature in RPN and explains its usefulness from the semantic perspective.

features and explanations. Figure 3 outlines the structure of our prompt. The descriptions of the dataset and attributes provide contextual information for the LLM to understand the data and apply domain knowledge. The value ranges of attributes are useful for selecting appropriate feature transformations, e.g., min-max normalization when the scale is too large. We include the descriptions of transformation operators as they help the LLM parse example features in RPN syntax and construct syntactically valid feature strings. The output template not only structures the output but instructs the LLM to reason about the usefulness of the constructed features and offer semantic explanations, utilizing the chain-of-thought technique (Wei et al., 2022; Kojima et al., 2022). We additionally add a constraint instruction to use no more than a certain number of operators, which reduces the search space and regularizes the solutions. Figure 4 shows an example LLM output. The prompt may further include dataset statistics like mean, standard deviation, and skewness of the attributes.

We initialize the prompt with $k$ random features from the constrained feature space $\tilde{\boldsymbol{x}}_1, \ldots, \tilde{\boldsymbol{x}}_k \in \mathbb{X}_{\mathbb{T}}^{(2)}$ represented in cRPN for demonstration, where the feature attributes are sampled per the softmax probabilities of feature importance by fitting the downstream model on the training data. This lets the LLM start search from a small feature space where it is easier to identify the basic patterns of promising features. Optionally, we can import external example features. We prompt the LLM to construct a fixed number of $m$ new feature in an iteration. For each constructed feature string $f$, we first try to obtain the cRPN expression $f^c$ to check whether $f^c$ is syntactically valid and not a duplicate of candidate features. If both criteria are met, we evaluate the performance score of adding the single feature to the dataset $s = \mathcal{E}_M(\mathbb{X} \cup \{\tilde{\boldsymbol{x}}_{f^c}\}, \boldsymbol{y})$ through cross validation on the training data and add $\langle f^c, s \rangle$ to the candidate set $\mathbb{F}_{cand}$. When $f^c$ is among the top-$k$ candidate features in terms of the score $s$, we update prompt examples with the top-$k$ pairs $\langle f', s' \rangle \in \mathbb{F}_{cand}$ ranked in the ascending order, taking score increment $s' - \mathcal{E}_M(\mathbb{X}, \boldsymbol{y})$ from the baseline. We also provide the feedback of previously constructed features with scores or error messages for improvement. We then instruct the LLM to construct additional features using the updated prompt. To select candidate features,

---

**Algorithm 1:** AutoFE by Prompting

**Input** : Dataset $D = \langle \mathbb{X}, \boldsymbol{y} \rangle$, downstream model $M$, large language model $LLM$, and optionally an external set of features with evaluation scores $\mathbb{F}_{ext}$

**Output:** A set of engineered features $\mathbb{F}$

1 Initialize prompt $P$ with dataset descriptions and example features; $\mathbb{F}_{cand} \leftarrow \mathbb{F}_{ext}$ if $\mathbb{F}_{ext}$ is available, otherwise $\mathbb{F}_{cand} \leftarrow \emptyset$; $\mathbb{F}_{set} \leftarrow \emptyset$

2 **repeat**

3     $\mathbb{F}_{LLM} = \{f_1, \ldots, f_m\} \leftarrow LLM(P)$        ▷ Feature generation

4     **for** each $f \in \mathbb{F}_{LLM}$ **do**

5        $f^c \leftarrow$ Canonicalize $f$

6        **if** $f^c$ is valid and $f^c \notin \mathbb{F}_{cand}$ **then**        ▷ Feature evaluation

7           Evaluate cross validation performance $s \leftarrow \mathcal{E}_M(\mathbb{X} \cup \{\boldsymbol{x}_{f^c}\}, \boldsymbol{y})$

8           $\mathbb{F}_{cand} \leftarrow \mathbb{F}_{cand} \cup \{\langle f^c, s \rangle\}$

9        **end**

10     **end**

11     Update $P$ such that $P$ contains the top-$k$ $\langle f', s' \rangle \in \mathbb{F}_{cand}$ as ordered by $s'$

12     **if** feature selection **then**

13        **for** $n \leftarrow 1$ **to** $|\mathbb{F}_{cand}|$ **do**        ▷ Feature selection

14           $\mathbb{F}_n \leftarrow$ The top-$n$ features in $\mathbb{F}_{cand}$ as ordered by $s$

15           Evaluate validation performance $s_n \leftarrow \mathcal{E}_M(\mathbb{X} \cup \check{\mathbb{X}}_{\mathbb{F}_n}, \boldsymbol{y})$

16        **end**

17        $\mathbb{F}_{set} \leftarrow \mathbb{F}_{set} \cup \{\langle \mathbb{F}_{n^*}, s_{n^*} \rangle\}$, with $n^* \leftarrow \arg\max_n s_n$

18     **end**

19 **until** stopping criteria are met

20 **return** $\mathbb{F}$ in $\mathbb{F}_{set}$ with the maximum validation score

---

we successively add candidate features to the dataset from the best to the worst and determine the optimal number of features to add based on validation performance, which is evaluated over sets of candidate features and thus takes feature interactions into account.

Algorithm 1 summarizes our methodology. The size of the prompt scales linearly with the number of features in the dataset $d$ and the number of example features $k$ and stays roughly constant across feature construction iterations. Thus, the cost of an LLM generation step in line 3 is almost constant. The computation cost of feature evaluation in line 7 is also constant, preserving the efficiency and scalability of our algorithm. The evaluations in line 7 and at lines 13-16 are parallelizable.

In our algorithm, the LLM is instructed to perform as a problem solver (Yang et al., 2024). Analogous to evolutionary algorithms that generate new solutions through crossover and mutations on high-fitness candidates (Smith & Bull, 2005; Zhu et al., 2022a; Morris et al., 2024), we provide top-performing features in the prompt. By learning examples and scores in-context (Brown et al., 2020), the LLM recognizes the patterns of promising features and generates new features that are likely to be useful. It can make analogies to, modify, or combine example features in the prompt (Appendix F.3). Early in the search, we expect greater exploration due to the diversity of initial examples. As iterations progress, the LLM exploits promising feature spaces more, gradually refining the search until convergence. The dataset semantic information enhances the effectiveness of feature search through the guidance as a prior. The LLM's temperature can be adjusted to balance exploration and exploitation, with higher temperatures encouraging more diverse solutions and lower temperatures favoring incremental changes to example features.

We adopt the same set of transformation operators $\mathbb{T}$ as those in (Zhu et al., 2022b), including:

- Unary transformations: logarithm, reciprocal, square root, and min-max normalization;

- Binary transformations: addition, subtraction, multiplication, division, and modulo.

In min-max normalization, we take the statistics from the training data. Other transformations require only the information of a single instance. Hence, all transformations can be performed on an individual test instance without leaking other instances' information. Data leakage (Overman et al., 2024) is an issue that has not been properly addressed in many existing AutoFE works.

## 5 EXPERIMENTS

### 5.1 EXPERIMENTAL SETUP

We benchmark performance on public real-world datasets from Kaggle and UCI repositories covering different domains (Appendix D.1). The descriptive information of datasets and attributes is retrieved from the sources without further processing. The downstream models we evaluate include linear models (LASSO for regression tasks and logistic regression for classification tasks), Random

**Table 1:** Comparison of overall performance. For each compared method, the left and right columns show the performance without and with post AutoFE parameter tuning of downstream model algorithms, respectively. The best results are boldfaced, and the second best results are underlined.

| Model | Raw | DIFER | | OpenFE | | CAAFE | | | | OCTree | | PromptFE (ours) | | | |
|---|---|---|---|---|---|---|---|---|---|---|---|---|---|---|---|
| | | | | | | GPT-3.5 | | GPT-4 | | GPT-4 | | GPT-3.5 | | GPT-4 | |
| Linear Model | 0.5636 | 0.6248 | 0.6369 | 0.5871 | 0.5866 | 0.5946 | 0.5941 | 0.5945 | 0.5946 | 0.6038 | 0.6044 | 0.6485 | 0.6487 | **0.6532** | 0.6526 |
| | 14.00 | 9.17 | 5.83 | 10.58 | 9.92 | 10.00 | 9.50 | 10.50 | 9.83 | 8.25 | 7.33 | 5.08 | 3.50 | 3.33 | **3.17** |
| Random Forests | 0.7252 | 0.7400 | 0.7411 | 0.7380 | 0.7376 | 0.7387 | 0.7378 | 0.7357 | 0.7352 | 0.7348 | 0.7346 | 0.7408 | **0.7412** | 0.7392 | 0.7393 |
| | 12.71 | 8.29 | 5.86 | 8.07 | 8.64 | 6.29 | 7.50 | 9.00 | 10.79 | 9.14 | 10.79 | **4.43** | 4.57 | 6.29 | 7.64 |
| Light-GBM | 0.7364 | 0.7504 | 0.7531 | 0.7454 | 0.7476 | 0.7457 | 0.7461 | 0.7405 | 0.7457 | 0.7409 | 0.7403 | 0.7522 | **0.7558** | 0.7542 | 0.7538 |
| | 10.43 | 8.86 | 6.29 | 9.50 | 9.14 | 9.07 | 8.14 | 9.00 | 8.50 | 11.21 | 12.21 | 5.71 | **3.57** | 3.57 | 4.79 |
| Mean | 0.6806 | 0.7091 | 0.7140 | 0.6953 | 0.6958 | 0.6979 | 0.6976 | 0.6950 | 0.6967 | 0.6976 | 0.6975 | 0.7171 | 0.7185 | **0.7187** | 0.7183 |
| Mean Rank | 12.30 | 8.75 | 6.00 | 9.33 | 9.20 | 8.38 | 8.33 | 9.45 | 9.70 | 9.60 | 10.25 | 5.08 | **3.90** | 4.45 | 5.30 |

Forests (Breiman, 2001), and LightGBM (Ke et al., 2017). For linear models, we target-encode categorical features and min-max scale all features. We tune downstream model algorithm parameters by randomized search prior to and post AutoFE, because the model may need reconfiguration to accommodate the added features. Data are randomly split into training (64%), validation (16%), and test (20%) sets. We evaluate regression performance with $1 - (relative\ absolute\ error)$[1] and classification performance with accuracy. A higher evaluation score indicates better performance.

We compare PromptFE with the following state-of-the-art AutoFE methods: (1) DIFER (Zhu et al., 2022b): A neural network-based method that optimizes features in the embedding space using LSTMs to encode and decode features; (2) OpenFE (Zhang et al., 2023): An expansion-reduction method that evaluates features up to a certain order using a feature boost algorithm; (3) CAAFE (Hollmann et al., 2023): An LLM-based method that produces Python code to manipulate datasets stored in Pandas data frames; (4) OCTree (Nam et al., 2024): An LLM-based method that generates rules to manipulate datasets and encodes features using decision tree algorithms.

We employ `gpt-3.5-turbo-0125`[2] and `gpt-4-0613`[2] as the LLMs. For PromptFE, we include $k = 10$ example features in the prompt and set the temperature of LLMs to 1 based on validation. We instruct the LLM to construct $m = 1$ feature in each generation step for the best control of feature generation. We perform feature selection each time 10 new candidate features are constructed and terminate the algorithm once we have 200 candidate features. Parameters of the baseline methods are initialized per the corresponding papers. We make five repeated runs.

## 5.2 Performance Comparison

Table 1 compares the overall performance between PromptFE and the baseline methods. Full results are presented in Appendix D.5[3]. PromptFE attains the best mean performance score and the lowest mean rank for all three downstream models, yielding over 5% mean performance gain and over 15% gain for linear models. We observe the greatest gain for linear models because unlike Random Forests and LightGBM, they cannot learn non-linear relationships thenselves. The performance margin between PromptFE and baselines other than DIFER is statistically significant with $p < 0.01$ by Friedman-Nemenyi test. PromptFE consistently outperforms CAAFE and OCTree, showing the robustness of PromptFE that reduces the search space with pre-defined operators and represents features in compact cRPN. Post-AutoFE parameter tuning brings the greatest performance improvement to DIFER, as it adds the most features to datasets (Appendix D.9). Compared with DIFER evaluating over 2000 candidate features during feature search, PromptFE evaluates only 200 candidate features (Appendix D.10). The higher efficiency of PromptFE is brought by the construction of semantically meaningful and effective features with the guidance of dataset semantic information.

We note that in PromptFE, using GPT-4 yields better performance for linear models but slightly worse performance for Random Forests than GPT-3.5. We speculate this is because the stronger in-context learning capability of GPT-4 increases the tendency of overfitting example features. One way to address this is to include more example features in the prompt to fully leverage GPT-4's enhanced in-context learning capability (Appendix D.8).

---

[1] $1 - \frac{\sum_i |y_i - \hat{y}_i|}{\sum_i |y_i - \bar{y}|}$, $y$ is the target and $\hat{y}$ is the prediction.

[2] https://platform.openai.com/docs/models

[3] We were unable to complete the OCTree evaluations using GPT-3.5 as it easily got stuck with iteratively generating rules that triggered errors in our experiments.

**Table 2:** Comparison of PromptFE with ablated versions. For each compared version, the left and middle columns show the performance without and with post AutoFE parameter tuning of downstream model algorithms, respectively, and the right column shows the number of LLM generations. Statistical significance of performance difference by Friedman-Nemenyi test is indicated with * for $p < 0.05$ and ** for $p < 0.01$.

| | Model | w/o Semantic Context | | | w/o Canonicalization | | | PromptFE | | |
|---|---|---|---|---|---|---|---|---|---|---|
| GPT-3.5 | Linear Model | 0.6411 | 0.6433 | 443.4 | 0.6471 | 0.6486 | 349.1 | 0.6485 | 0.6487 | 356.7 |
| | Random Forests | 0.7326** | 0.7328** | 472.5 | 0.7372 | 0.7373 | 358.0 | 0.7408 | 0.7412 | 370.4 |
| | LightGBM | 0.7479* | 0.7494 | 490.0 | 0.7485 | 0.7490 | 348.9 | 0.7522 | 0.7558 | 360.2 |
| | Mean | 0.7105** | 0.7118** | 469.9 | 0.7141 | 0.7148 | 352.2 | 0.7171 | 0.7185 | 362.7 |
| GPT-4 | Linear Model | 0.6437 | 0.6461 | 253.9 | 0.6462 | 0.6463 | 323.6 | 0.6532 | 0.6526 | 326.3 |
| | Random Forests | 0.7285* | 0.7288* | 262.9 | 0.7366 | 0.7366 | 315.7 | 0.7392 | 0.7393 | 333.0 |
| | LightGBM | 0.7420** | 0.7437 | 250.7 | 0.7461* | 0.7480 | 328.5 | 0.7542 | 0.7538 | 335.7 |
| | Mean | 0.7078** | 0.7092** | 255.9 | 0.7128** | 0.7135* | 322.5 | 0.7187 | 0.7183 | 331.9 |

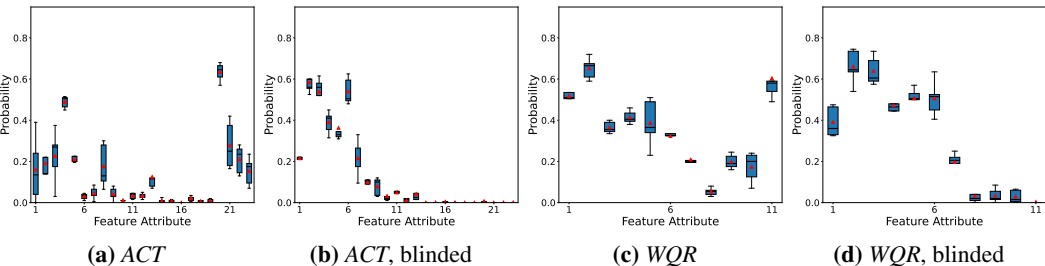

**(a)** *ACT*      **(b)** *ACT*, blinded      **(c)** *WQR*      **(d)** *WQR*, blinded

**Figure 6:** Distributions of feature attribute selection in the constructed features for linear models with GPT-4.

## 5.3 EFFECT OF SEMANTIC CONTEXT

We compare with the blinded version without dataset semantic information (Appendix C.2). From Table 2, PromptFE outperforms the blinded version for all downstream models with statistical significance. The performance difference is more pronounced for Random Forests and LightGBM, likely because the inclusion of non-semantically meaningful features by the blinded version consumes model capacity and causes greater overfitting to the training data. GPT-4 constructs features more efficiently than GPT-3.5 due to stronger capabilities. The incorporation of dataset semantic context improves the feature construction efficiency of GPT-3.5 but reduces that of GPT-4, as it guides to more focused feature spaces that increase the chances of duplication with candidate features.

## 5.4 EFFECT OF FEATURE EXPRESSION CANONICALIZATION

We compare with the ablated version without canonicalization of feature expressions. From Table 2, PromptFE outperforms the ablated version for all downstream models. Without canonicalization, we observe a slight decrease in the number of LLM generations. Since a feature can be represented in different expressions, the chances of duplication with the expressions of candidate features during feature search are reduced. However, the effectiveness of the features constructed by the LLM degrades in this setting due to increased difficulty in learning optimal feature patterns.

## 5.5 FEATURE ATTRIBUTE SELECTION

Figure 6 shows the distributions of feature attributes in the constructed features for linear models using GPT-4. Without semantic context, the LLM tends to prioritize earlier feature attributes in the dataset while paying less attention to later ones. In comparison, PromptFE is informed by the semantic context. Specifically, Attribute 20 *CD4 at baseline* in *ACT* and Attribute 11 *alcohol* in *WQR*, which contain critical information for predicting the respective targets *censoring indicator* and *quality*, are consistently among the most frequent ones. This illustrates how the LLM leverages dataset semantic information to construct semantically meaningful and effective features in PromptFE.

## 5.6 PERFORMANCE ANALYSIS

We study the performance for linear models with GPT-3.5 from ten repeated runs. Figures 7-10 display the slope and $p$-value from one-tailed t-tests in OLS regressions, with the shaded area showing one standard deviation above and below the mean curve.

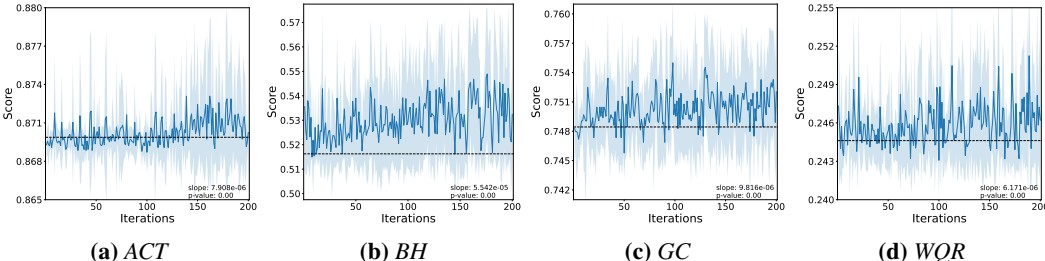

**(a)** *ACT*     **(b)** *BH*     **(c)** *GC*     **(d)** *WQR*

**Figure 7:** The cross validation score of candidate features on training data across iterations. The baseline cross validation score with raw dataset features is indicated with the dash line.

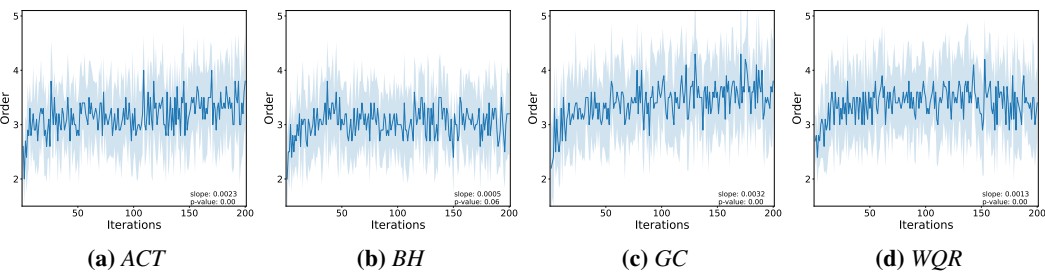

**(a)** *ACT*     **(b)** *BH*     **(c)** *GC*     **(d)** *WQR*

**Figure 8:** The order of candidate features across iterations.

**Feature Learning.** We examine the cross validation score of candidate features across iterations. Figure 7 shows a significantly upward trend in the score, with most constructed features improving the performance. This demonstrates that PromptFE effectively improves the quality of constructed features through in-context learning of top-performing examples during feature search.

**Feature Complexity.** We examine the order of candidate features across iterations. Figure 8 shows that the feature order increases rapidly in early iterations and stabilizes over time. PromptFE effectively constructs complex features within promising feature spaces. Moreover, our constraint instruction offers regularization that prevents the construction of overly complex features.

**Feature Divergence.** We analyze the divergence of a new candidate feature from previous ones during feature search. We compute the edit distance between canonical feature expression trees using the algorithm by Zhang & Shasha (1989) and normalize the distance by the total number of nodes in both trees. Figure 9 shows the mean normalized tree edit distance between the current candidate feature and the previous five features across iterations. The observed downward trend indicates that feature search converges over iterations.

**Feature Construction Efficiency.** We examine the number of LLM generations needed to construct new candidate features across iterations. Figure 10 shows a slightly upward trend in the number of LLM generations, due to increasing difficulty of constructing non-duplicate features and higher likelihood of producing syntactical errors as features become more complex. Since the increase is non-significant, PromptFE remains scalable to a large number of iterations.

## 5.7 HYPERPARAMETER EFFECT

**Number of Examples in Prompt.** Table 3 reports the maximum validation score across iterations along with the number of LLM generations by varying the number of example features provided

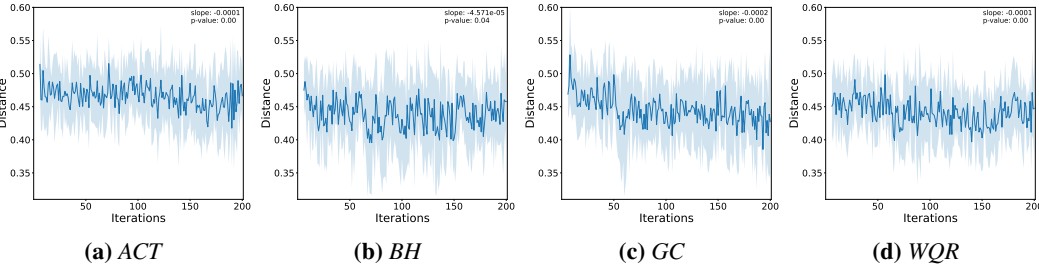

**(a)** *ACT*     **(b)** *BH*     **(c)** *GC*     **(d)** *WQR*

**Figure 9:** The mean normalized edit distance between a candidate feature and previous five features.

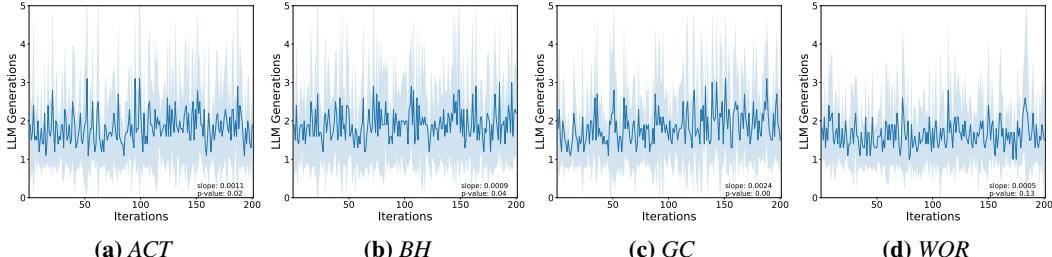

**(a)** *ACT*  **(b)** *BH*  **(c)** *GC*  **(d)** *WQR*

**Figure 10:** The number of LLM generations to construct a new candidate feature across iterations.

**Table 3:** Effect of the number of example features in the prompt with GPT-3.5. For each compared setting, the left column shows the validation score, and the right column shows the number of LLM generations.

| Model | Dataset | Number of Examples | | | | | | | |
| --- | --- | --- | --- | --- | --- | --- | --- | --- | --- |
| | | 1 | | 5 | | 10 | | 20 | |
| RF | AF | 0.7895 | 507.2 | **0.7930** | 409.2 | 0.7914 | 393.2 | 0.7860 | **372.8** |
| | WQR | 0.3897 | 339.4 | 0.3937 | 329.8 | **0.3948** | 362.6 | 0.3940 | **330.0** |
| | CD | 0.8212 | 480.2 | 0.8213 | 371.2 | **0.8219** | 349.8 | 0.8218 | **343.2** |
| LGBM | AF | 0.8421 | 440.4 | **0.8433** | 404.6 | 0.8430 | **380.2** | 0.8420 | 384.2 |
| | WQR | 0.4248 | 346.0 | 0.4294 | 334.8 | 0.4301 | **322.8** | **0.4333** | 330.4 |
| | CD | **0.8228** | 449.4 | 0.8224 | 361.2 | 0.8226 | 352.2 | 0.8228 | **321.2** |
| Mean | | 0.6817 | 427.1 | 0.6839 | 368.5 | **0.6840** | 360.1 | 0.6833 | **347.0** |

in the prompt. We observe that the best performance is achieved with 10 examples. Additionally, feature construction efficiency improves as the number of examples increases, as more examples can help the LLM reduce errors and generate more diverse features. Nonetheless, too many examples hinder the in-context learning of optimal feature patterns, as shown by the performance decline. The performance difference with 10 examples and with 1 example is statistically significant with $p < 0.05$ by one-tailed paired t-tests.

**Temperature.** Table 4 reports the maximum validation score across iterations with the number of LLM generations under different LLM temperatures. We observe the best performance and efficiency when the temperature is around 1. Lower temperatures increase the likelihood of the LLM repeating previously constructed features, while higher temperatures make

**Table 4:** Effect of the LLM temperature with GPT-3.5.

| Model | Dataset | Temperature | | | | | |
| --- | --- | --- | --- | --- | --- | --- | --- |
| | | 0.5 | | 1 | | 1.5 | |
| RF | AF | 0.7875 | 794.4 | 0.7914 | **393.2** | **0.7916** | 609.2 |
| | CD | 0.8211 | 823.2 | **0.8219** | **349.8** | 0.8218 | 672.6 |
| LGBM | AF | 0.8365 | 1313.2 | **0.8430** | **380.2** | 0.8418 | 627.6 |
| | CD | 0.8225 | 519.8 | **0.8226** | **352.2** | 0.8223 | 662.6 |
| Mean | | 0.8169 | 862.7 | **0.8197** | **368.9** | 0.8194 | 643.0 |

the LLM more prone to producing errors in the generations, both reducing feature construction efficiency. A temperature at 1 provides the best tradeoff between exploration and exploitation in feature search. The performance difference with the temperature at 1 and at 0.5 is statistically significant with $p < 0.01$ by one-tailed paired t-tests.

# 6 CONCLUSION

In this paper, we present a novel LLM-based AutoFE framework for effective, efficient, and interpretable feature engineering that leverages the semantic information of datasets. It features an elegant approach to instructing the LLM to generate semantically meaningful features with explanations by providing dataset descriptions and example features in cRPN expressions. The LLM iteratively explores the feature space and improves feature construction by learning top-performing examples in context. We have demonstrated in extensive experiments that our approach significantly outperforms state-of-the-art AutoFE methods. The incorporation of semantic context from dataset descriptions and the proposed feature canonicalization scheme both contribute to performance improvement. We have also provided comprehensive analysis on the LLM-based feature construction process. Our work opens up new possibilities for further LLM-driven applications on automated machine learning methodologies and underscores the potential of utilizing semantic information. A future direction is to introduce adaptive methods for prompt design.

## ETHICAL STATEMENT

All datasets used in this work are publicly available, free of personal information, and intended for research purposes only. Our use of GPT models complies with the terms and conditions of OpenAI.

## REPRODUCIBILITY STATEMENT

The anonymized source code of this work can be accessed at `https://anonymous.4open. science/r/PromptFE_share-8F26`.

## USE OF LARGE LANGUAGE MODELS

Large language models are not used in paper writing.

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

## A   DISCUSSION ON CANONICAL RPN FEATURE REPRESENTATION

### A.1   WHY RPN

RPN (Hamblin, 1962) provides a compact and unambiguous form of feature representation. In contrast, infix expression requires extra information such as brackets to determine operator precedence. Without brackets, the feature in infix expression col-0 − ( col-1 + col-2 ) would be indistinguishable from the feature ( col-0 − col-1 ) + col-2, while both features are distinctively encoded in RPN. Such compactness and unambiguity of RPN facilitate sequential modeling since there is no need to model the extra information, e.g., the positions of brackets.

Compared with other forms of feature representation such as prefix expression of depth-first traversal or breadth-first traversal, RPN better encodes the recursive structure of the expression tree. The bottom-up enumeration of tree nodes makes it easy for the LLM to evaluate the feature expression by scanning the sequence from left to right, for instance, ((col-0 col-1 −) col-2 +) (parentheses denote recursion). Using the prefix expression (+ (− col-0 col-1) col-2) or breadth-first expression (+ (− [col-2] col-0 col-1)), however, the LLM always needs to look back to find the operator, which undermines sequential modeling. We find in our experiments that when switching to prefix feature expressions, the LLM encounters difficulty in generating syntactically valid feature expressions.

### A.2   WHY CANONICALIZATION

Although there is one-to-one mapping between feature expression trees and RPN expressions, a feature that contains commutative operators (like addition and multiplication) can be represented by different RPN expressions, since the child nodes of these operators are unordered. We introduce a canonicalization scheme: arranging operator nodes before feature nodes and lexicographically sorting the nodes within each group. Through canonicalization, we create one-to-one mapping between features and cRPN expressions. This ensures the consistency of our feature representations and facilitates the in-context learning of feature patterns.

By arranging operator nodes before feature nodes, we also introduce left skewness to the expression tree that enhances the clarity of the recursive structure in cRPN. As illustrated in Figure 2, the original feature expression (col-2 (col-1 col-0 +) ∗) becomes ((col-0 col-1 +) col-2 ∗) after canonicalization, so that the LLM does not need to look back for col-2 when evaluating the expression.

## B   CONVERSION BETWEEN FEATURE EXPRESSION TREE AND RPN

Algorithms 2 and 3 detail the process of conversion between a feature expression tree and an RPN feature string. We check the RPN syntactical validity of a feature string in Algorithm 3 by checking whether there is enough child node in the stack in line 6 and the size of the stack is exactly one (the root) in line 13 returning the output.

---

**Algorithm 2:** Feature Expression Tree to RPN

**Input** : A feature expression tree $T$
**Output:** An RPN feature string $f$

1   $r \leftarrow$ the root of $T$
2   Initialize string $f \leftarrow \epsilon$, stack $S \leftarrow [r]$, and $visited \leftarrow \emptyset$
3   **repeat**
4     $u \leftarrow S.peek()$
5     **if** $u \in visited$ **then**
6       $f.append(u)$
7       $S.pop()$
8     **end**
9     **else**
10       **for** each child $v$ of $u$ in the reverse order **do**
11         $S.push(v)$
12       **end**
13       $visited \leftarrow visited \cup \{u\}$
14     **end**
15 **until** $S$ is empty
16 **return** $f$

---

**Algorithm 3:** RPN to Feature Expression Tree

**Input** : An RPN feature string $f$
**Output:** The root of a feature expression tree $T$

1   Initialize stack $S \leftarrow []$
2   **for** $i \leftarrow 1$ **to** $|f|$ **do**
3     $u \leftarrow$ the $i$-th element of $f$
4     **if** $u$ is an operator **then**
5       $o \leftarrow$ the arity of $u$
6       **for** $j \leftarrow 1$ **to** $o$ **do**
7         $v \leftarrow S.pop()$
8         Prepend $v$ to the list of children of $u$
9       **end**
10     **end**
11     $S.push(u)$
12 **end**
13 **return** $S.pop()$

---

# C   EXAMPLE PROMPTS

## C.1   COMPLETE PROMPT

Figure 11 shows an example of complete prompts used in our main experiments.

---

Figure 11: Example complete prompt on the Credit Default dataset.

Dataset description:
This dataset contains information on default payments, demographic factors, credit data, history of payment, and bill statements of credit card clients in Taiwan from April 2005 to September 2005. Dataset contains the following columns:
col-0 (int) [10000, 800000]: LIMIT_BAL: Amount of given credit in NT dollars (includes individual and family/supplementary credit
col-1 (category) {1, 2}: SEX: Gender (1=male, 2=female)
col-2 (category) {0, 1, 2, 3, 4, 5, 6}: EDUCATION: (1=graduate school, 2=university, 3=high school, 4=others, 5=unknown, 6=unknown)
col-3 (category) {0, 1, 2, 3}: MARRIAGE: Marital status (1=married, 2=single, 3=others)
col-4 (int) [21, 79]: AGE: Age in years
col-5 (category) {-2, -1, 0, 1, 2, 3, 4, 5, 6, 7, 8}: PAY_0: Repayment status in September, 2005 (-1=pay duly, 1=payment delay for one month, 2=payment delay for two months, . . . 8=payment delay for eight months, 9=payment delay for nine months and above)
. . .

---

col-23 (category) {0, 1}: default.payment.next.month: Default payment (1=yes, 0=no)
We have the following unary operators:
log: element-wise logarithm of the absolute value
sqrt_abs: element-wise square root of the absolute value
min_max: element-wise min-max normalization
reciprocal: element-wise reciprocal
We have the following binary operators:
+: element-wise addition of two columns
−: element-wise subtraction of two columns
∗: element-wise multiplication of two columns
/: element-wise division of two columns
mod_column: element-wise modulo of two columns
Feature strings are reverse Polish notation (RPN) expressions that operate on the columns of our dataset. Each feature string constructs an extra column that is useful for the downstream model Random Forests to predict the target col-23. The model will be trained on the dataset with the constructed columns and evaluated on a holdout set. The best columns will be selected.
Below are feature strings arranged in ascending order based on their performance scores. Higher scores are better.

Feature
col-17,col-21,*,col-20,+,sqrt_abs
Score
0.0011
. . .
Feature
col-4,col-6,*,col-12,col-16,-,sqrt_abs,*
Score
0.0014

Previous feature:
col-4,col-5,col-6,+,col-12,col-16,-,*
Error: invalid RPN expression

Give me a new feature string that is different from all strings above and has a higher score. Use no more than five operators. Make sure all columns and operators exist and do not include the target column. Follow the syntax of RPN.

Output format:
Feature

(Feature name and description)

Usefulness
(Explanation why this adds useful real world knowledge to predict the target col-23 according to dataset description)

## C.2 SEMANTICALLY BLINDED PROMPT

Figure 12 shows an example of semantically blinded prompts used in our experiments in Section 5.3.

Figure 12: Example semantically blinded prompt on the Credit Default dataset.

Dataset contains the following columns:
col-0
col-1
col-2
col-3
col-4
col-5
. . .
col-23
We have the following unary operators:
log: element-wise logarithm of the absolute value

sqrt_abs: element-wise square root of the absolute value
min_max: element-wise min-max normalization
reciprocal: element-wise reciprocal
We have the following binary operators:
+: element-wise addition of two columns
−: element-wise subtraction of two columns
∗: element-wise multiplication of two columns
/: element-wise division of two columns
mod_column: element-wise modulo of two columns
Feature strings are reverse Polish notation (RPN) expressions that operate on the columns of our dataset. Each feature string constructs an extra column that is useful for the downstream model Random Forests to predict the target col-23. The model will be trained on the dataset with the constructed columns and evaluated on a holdout set. The best columns will be selected.
Below are feature strings arranged in ascending order based on their performance scores. Higher scores are better.

Feature
col-17,col-21,*,col-20,+,sqrt_abs
Score
0.0011
. . .
Feature
col-4,col-6,*,col-12,col-16,-,sqrt_abs,*
Score
0.0014

Previous feature:
col-4,col-6,*,col-12,col-16,-,sqrt_abs,*
Error: duplication with candidate features

Give me a new feature string that is different from all strings above and has a higher score. Use no more than five operators. Make sure all columns and operators exist and do not include the target column. Follow the syntax of RPN.

Output format:
Feature

(Feature name and description)

Usefulness
(Explanation why this adds useful real world knowledge to predict the target col-23 according to dataset description)

# D    EXPERIMENTAL DETAILS

## D.1    DATASETS

Tables 5 and 6 summarize the statistics and sources of datasets used in our experiments. Datasets are selected such that they cover different domains and both regression and classification tasks, most of which have been used in previous works (Zhu et al., 2022a;b; Zhang et al., 2023; Hollmann et al., 2023).

**Table 5:** Statistics of datasets. The datasets cover different domains and tasks and vary in sizes.

| Name | Task | # Samples | # Features | # Numerical | # Categorical |
|---|---|---|---|---|---|
| Airfoil (AF) | Regression | 1,503 | 5 | 5 | 0 |
| Boston Housing (BH) | Regression | 506 | 13 | 12 | 1 |
| Bikeshare (BS) | Regression | 731 | 10 | 6 | 4 |
| Wine Quality Red (WQR) | Regression | 1,599 | 11 | 11 | 0 |
| AIDS Clinical Trials (ACT) | Classification | 2,139 | 23 | 9 | 14 |
| Credit Default (CD) | Classification | 30,000 | 23 | 14 | 9 |
| German Credit (GC) | Classification | 1,000 | 20 | 10 | 10 |

**Table 6:** Sources of datasets. The descriptive information of datasets and attributes is retrieved from the sources without further processing.

| Name | Source |
|---|---|
| Airfoil (AF) | https://archive.ics.uci.edu/dataset/291/airfoil+self+noise |
| Boston Housing (BH) | https://www.kaggle.com/datasets/arunjangir245/boston-housing-dataset |
| Bikeshare (BS) | https://www.kaggle.com/datasets/marklvl/bike-sharing-dataset |
| Wine Quality Red (WQR) | https://archive.ics.uci.edu/dataset/186/wine+quality |
| AIDS Clinical Trials (ACT) | https://archive.ics.uci.edu/dataset/890/aids+clinical+trials+group+study+175 |
| Credit Default (CD) | https://www.kaggle.com/datasets/uciml/default-of-credit-card-clients-dataset |
| German Credit (GC) | https://archive.ics.uci.edu/dataset/573/south+german+credit+update |

## D.2    EXPERIMENTAL PLATFORM

All experiments are conducted on the Ubuntu 22.04.4 LTS operating system, 16 Intel(R) Core(TM) i7-7820X CPUs, and 4 NVIDIA GeForce RTX 2080 Ti GPUs, with the framework of Python 3.11.9 and PyTorch 1.12.1.

## D.3    FEATURE TRANSFORMATION OPERATORS

We list the details of all feature transformation operators below.

Unary transformations:

- Logarithm: Element-wise logarithm of the absolute value;
- Reciprocal: Element-wise reciprocal;
- Square root: Element-wise square root of the absolute value;
- Min-max normalization: Element-wise min-max normalization using the min and max values from the training data.

Binary transformations:

- Addition: Element-wise addition;
- Subtraction: Element-wise subtraction;
- Multiplication: Element-wise multiplication;
- Division: Element-wise division;
- Modulo: Element-wise modulo.

### D.4 Parameter Tuning of Downstream Models

We tune the parameters of downstream models prior to and post AutoFE using randomized search implemented in an `Sklearn` package[4]. Table 7 lists the configurations of parameter tuning for each downstream model. We set the number of randomized search iterations to 100.

**Table 7:** Hyperparameter search space for downstream models.

| Model | Parameter | Search Space[*] |
|---|---|---|
| Linear Model | regularization | loguniform(0.00001, 100) |
| Random Forests | num estimators | randint(5, 250) |
|  | max depth | randint(1, 250) |
|  | max features | uniform(0.01, 0.99) |
|  | max samples | uniform(0.1, 0.9) |
| LightGBM | num estimators | randint(10, 1000) |
|  | num leaves | randint(8, 64) |
|  | learning rate | loguniform(0.001, 1) |
|  | bagging fraction | uniform(0.1, 0.9) |
|  | feature fraction | uniform(0.1, 0.9) |
|  | reg lambda | loguniform(0.001, 100) |

[*] As defined in the `scipy.stats` documentation `https://docs.scipy.org/doc/scipy/reference/stats.html`.

### D.5 Full Results

Tables 8-10 detail the full experimental results corresponding to the results in Tables 1 and 2. Tables 11-15 report the sample standard deviations corresponding to the experimental results in Tables 8-10 and Tables 3-4, respectively.

### D.6 Relative Performance Improvement

Tables 16 and 17 report the percentage performance improvement of PromptFE over the baseline methods with GPT-3.5 and GPT-4, respectively, corresponding to the experimental results in Tables 1 and 8.

### D.7 Statistical Tests

We perform the Friedman test (Friedman, 1937) to determine whether there is statistically significant difference among the compared AutoFE methods. The Friedman test $p$-values for the results in Tables 1 and 2 are $4.26 \times 10^{-50}$ and $3.95 \times 10^{-34}$, respectively. Hence, we can reject the null hypothesis that the performance is the same for all methods. We perform the Nemenyi post-hoc test (Nemenyi, 1963) to further determine which AutoFE methods have different performance. Tables 18, 20, and 21 summarize the $p$-values for the pairwise comparisons in Tables 1 and 2. From Table 18, the performance difference between our method PromptFE and baseline methods other than DIFER (Zhu et al., 2022b) is statistically significant at the $p = 0.01$ level. From Table 20, the performance difference between the full version of PromptFE and the semantically blinded version is statistically significant at the $p = 0.01$ level. From Table 21, the performance difference is statistically significant at the $p = 0.05$ level for the cases with GPT-3.5 and post-AutoFE parameter tuning as well as GPT-4 without post-AutoFE parameter tuning.

To examine the performance difference when using Random Forests and LightGBM, we perform additional statistical tests for the results in Table 1 excluding the linear model results. The Friedman test $p$-value is $1.28 \times 10^{-25}$. Table 19 summarizes the $p$-values from the Nemenyi post-hoc test for pairwise comparison. We observe that PromptFE with GPT-3.5 and post-AutoFE parameter

---

[4] `https://scikit-learn.org/1.5/modules/generated/sklearn.model_selection.RandomizedSearchCV.html`

**Table 8:** Full experimental results of Table 1 comparison of overall performance. For each compared method, the left and right columns show the results without and with parameter tuning of the downstream model algorithm post AutoFE, respectively. The best results are boldfaced, and the second best results are underlined.

| Model | Dataset | Raw | DIFER | | OpenFE | | CAAFE | | | | OCTree | | PromptFE (ours) | | | |
|---|---|---|---|---|---|---|---|---|---|---|---|---|---|---|---|---|
| | | | | | | | GPT-3.5 | | GPT-4 | | GPT-4 | | GPT-3.5 | | GPT-4 | |
| Linear Model | AF | 0.3474 | 0.5870 | 0.6090 | 0.4300 | 0.4303 | 0.4011 | 0.4016 | 0.4376 | 0.4378 | 0.4698 | 0.4698 | 0.6612 | 0.6616 | **0.6649** | 0.6647 |
| | BH | 0.3776 | 0.5013 | 0.4994 | 0.3900 | 0.3880 | 0.4788 | 0.4765 | 0.4503 | 0.4506 | 0.4480 | 0.4545 | 0.4995 | 0.5025 | 0.5184 | **0.5289** |
| | BS | 1.0000 | – | – | – | – | – | – | – | – | – | – | – | – | – | – |
| | WQR | 0.2696 | 0.2475 | 0.2630 | 0.2713 | 0.2736 | 0.2742 | 0.2757 | 0.2776 | 0.2776 | 0.2774 | 0.2778 | 0.2722 | 0.2745 | 0.2713 | 0.2748 |
| | ACT | 0.8505 | 0.8715 | **0.8799** | 0.8729 | 0.8729 | 0.8519 | 0.8514 | 0.8565 | 0.8570 | 0.8724 | 0.8720 | 0.8729 | 0.8794 | 0.8766 | 0.8762 |
| | CD | 0.8267 | 0.8273 | 0.8280 | 0.8265 | 0.8268 | 0.8265 | 0.8267 | 0.8238 | 0.8238 | 0.8270 | 0.8272 | 0.8282 | 0.8282 | 0.8288 | **0.8288** |
| | GC | 0.7100 | 0.7140 | 0.7420 | 0.7320 | 0.7280 | 0.7350 | 0.7330 | 0.7210 | 0.7210 | 0.7280 | 0.7250 | 0.7570 | 0.7460 | **0.7590** | 0.7420 |
| Mean | | 0.5636 | 0.6248 | 0.6369 | 0.5871 | 0.5866 | 0.5946 | 0.5941 | 0.5945 | 0.5946 | 0.6038 | 0.6044 | 0.6485 | 0.6487 | **0.6532** | 0.6526 |
| Mean Rank | | 14.00 | 9.17 | 5.83 | 10.58 | 9.92 | 10.00 | 9.50 | 10.50 | 9.83 | 8.25 | 7.33 | 5.08 | 3.50 | 3.33 | **3.17** |
| Random Forests | AF | 0.7677 | 0.7650 | 0.7786 | 0.7579 | 0.7682 | 0.7711 | 0.7693 | 0.7696 | 0.7720 | 0.7603 | 0.7655 | 0.7709 | **0.7787** | 0.7681 | 0.7749 |
| | BH | 0.5426 | **0.5718** | 0.5701 | 0.5658 | 0.5620 | 0.5556 | 0.5556 | 0.5512 | 0.5492 | 0.5519 | 0.5497 | 0.5549 | 0.5533 | 0.5543 | 0.5522 |
| | BS | 0.9446 | 0.9865 | 0.9871 | 0.9901 | 0.9901 | 0.9916 | 0.9916 | 0.9818 | 0.9816 | **0.9924** | 0.9922 | 0.9873 | 0.9881 | 0.9845 | 0.9848 |
| | WQR | 0.3662 | 0.3838 | 0.3832 | 0.3753 | 0.3729 | 0.3718 | 0.3718 | 0.3693 | 0.3693 | 0.3655 | 0.3656 | 0.3862 | 0.3845 | 0.3810 | 0.3810 |
| | ACT | 0.8808 | 0.8897 | 0.8897 | 0.8832 | 0.8841 | 0.8827 | 0.8855 | 0.8827 | 0.8827 | 0.8864 | 0.8822 | **0.8925** | 0.8921 | 0.8893 | 0.8864 |
| | CD | 0.8293 | 0.8285 | 0.8291 | 0.8287 | 0.8285 | 0.8291 | 0.8289 | 0.8294 | 0.8287 | 0.8291 | 0.8289 | 0.8295 | 0.8294 | **0.8295** | 0.8276 |
| | GC | 0.7450 | 0.7550 | 0.7500 | 0.7650 | 0.7570 | **0.7690** | 0.7620 | 0.7660 | 0.7630 | 0.7580 | 0.7580 | 0.7640 | 0.7620 | 0.7680 | 0.7680 |
| Mean | | 0.7252 | 0.7400 | 0.7411 | 0.7380 | 0.7376 | 0.7387 | 0.7378 | 0.7357 | 0.7352 | 0.7348 | 0.7346 | 0.7408 | **0.7412** | 0.7392 | 0.7393 |
| Mean Rank | | 12.71 | 8.29 | 5.86 | 8.07 | 8.64 | 6.29 | 7.50 | 9.00 | 10.79 | 9.14 | 10.79 | **4.43** | 4.57 | 6.29 | 7.64 |
| Light-GBM | AF | 0.8375 | 0.8285 | 0.8411 | 0.8188 | 0.8244 | 0.8364 | 0.8348 | **0.8430** | 0.8426 | 0.8234 | 0.8190 | 0.8311 | 0.8392 | 0.8366 | 0.8395 |
| | BH | 0.5537 | 0.5607 | 0.5636 | **0.5693** | 0.5618 | 0.5540 | 0.5571 | 0.5478 | 0.5501 | 0.5442 | 0.5438 | 0.5619 | 0.5644 | 0.5642 | 0.5595 |
| | BS | 0.9429 | 0.9763 | 0.9786 | 0.9751 | 0.9797 | 0.9555 | 0.9565 | 0.9449 | 0.9487 | 0.9690 | 0.9731 | 0.9737 | 0.9754 | 0.9801 | **0.9813** |
| | WQR | 0.3825 | 0.4145 | **0.4182** | 0.3898 | 0.3884 | 0.4131 | 0.4035 | 0.3902 | 0.3952 | 0.3877 | 0.3834 | 0.4118 | 0.4171 | 0.4021 | 0.4042 |
| | ACT | 0.8832 | 0.8794 | 0.8827 | 0.8808 | 0.8799 | 0.8822 | 0.8860 | 0.8827 | 0.8818 | 0.8879 | 0.8822 | 0.8888 | **0.8925** | 0.8902 | 0.8925 |
| | CD | 0.8300 | 0.8283 | 0.8277 | 0.8293 | 0.8287 | 0.8296 | 0.8298 | 0.8301 | 0.8294 | 0.8293 | 0.8292 | 0.8301 | 0.8297 | **0.8303** | 0.8294 |
| | GC | 0.7250 | 0.7650 | 0.7600 | 0.7550 | 0.7700 | 0.7490 | 0.7550 | 0.7450 | 0.7720 | 0.7450 | 0.7510 | 0.7680 | 0.7720 | **0.7760** | 0.7700 |
| Mean | | 0.7364 | 0.7504 | 0.7531 | 0.7454 | 0.7476 | 0.7457 | 0.7461 | 0.7405 | 0.7457 | 0.7409 | 0.7403 | 0.7522 | **0.7558** | 0.7542 | 0.7538 |
| Mean Rank | | 10.43 | 8.86 | 6.29 | 9.50 | 9.14 | 9.07 | 8.14 | 9.00 | 8.50 | 11.21 | 12.21 | 5.71 | **3.57** | 3.57 | 4.79 |
| Mean | | 0.6806 | 0.7091 | 0.7140 | 0.6953 | 0.6958 | 0.6979 | 0.6976 | 0.6950 | 0.6967 | 0.6976 | 0.6975 | 0.7171 | 0.7185 | **0.7187** | 0.7183 |
| Mean Rank | | 12.30 | 8.75 | 6.00 | 9.33 | 9.20 | 8.38 | 8.33 | 9.45 | 9.70 | 9.60 | 10.25 | 5.08 | **3.90** | 4.45 | 5.30 |

**Table 9:** Full experimental results of Table 2 performance comparison of PromptFE with and without dataset semantic context. For each compared version, the left and middle columns show the results without and with parameter tuning of the downstream model algorithm post AutoFE, respectively, and the right column shows the number of LLM generations. The results where the full version outperforms the blinded version are boldfaced.

| Model | Dataset | Raw | GPT-3.5 | | | | | | GPT-4 | | | | | |
|---|---|---|---|---|---|---|---|---|---|---|---|---|---|---|
| | | | w/o Semantic Context | | | PromptFE | | | w/o Semantic Context | | | PromptFE | | |
| Linear Model | AF | 0.3474 | 0.6613 | 0.6602 | 450.0 | 0.6612 | **0.6616** | 339.8 | 0.6678 | 0.6672 | 275.0 | **0.6649** | **0.6647** | 371.4 |
| | BH | 0.3776 | 0.4678 | 0.4794 | 438.0 | **0.4995** | **0.5025** | 378.6 | 0.4869 | 0.4996 | 295.6 | **0.5184** | **0.5289** | 335.4 |
| | BS | 1.0000 | – | – | – | – | – | – | – | – | – | – | – | – |
| | WQR | 0.2696 | 0.2643 | 0.2733 | 442.8 | **0.2722** | **0.2745** | 328.4 | 0.2645 | 0.2702 | 244.6 | **0.2713** | **0.2748** | 312.6 |
| | ACT | 0.8505 | 0.8790 | 0.8799 | 442.8 | 0.8729 | **0.8794** | 372.2 | 0.8720 | 0.8729 | 238.8 | **0.8766** | **0.8762** | 377.4 |
| | CD | 0.8267 | 0.8283 | 0.8283 | 454.8 | 0.8282 | 0.8282 | 342.0 | 0.8282 | 0.8289 | 238.2 | **0.8288** | 0.8288 | 250.4 |
| | GC | 0.7100 | 0.7460 | 0.7390 | 432.2 | **0.7570** | **0.7460** | 379.0 | 0.7430 | 0.7410 | 231.2 | **0.7590** | **0.7420** | 310.6 |
| Mean | | 0.5636 | 0.6411 | 0.6433 | 443.4 | **0.6485** | **0.6487** | 356.7 | 0.6437 | 0.6461 | 253.9 | **0.6532** | **0.6526** | 326.3 |
| Random Forests | AF | 0.7677 | 0.7644 | 0.7743 | 425.2 | **0.7709** | **0.7787** | 393.2 | 0.7610 | 0.7690 | 274.2 | **0.7681** | **0.7749** | 314.2 |
| | BH | 0.5426 | 0.5483 | 0.5483 | 479.2 | **0.5549** | **0.5533** | 374.4 | 0.5507 | 0.5491 | 238.4 | **0.5543** | **0.5522** | 278.6 |
| | BS | 0.9446 | 0.9628 | 0.9628 | 510.0 | **0.9873** | **0.9881** | 386.8 | 0.9535 | 0.9543 | 247.4 | **0.9845** | **0.9848** | 255.0 |
| | WQR | 0.3662 | 0.3749 | 0.3738 | 461.4 | **0.3862** | **0.3845** | 362.6 | 0.3666 | 0.3674 | 253.0 | **0.3810** | **0.3810** | 283.2 |
| | ACT | 0.8808 | 0.8864 | 0.8841 | 475.8 | **0.8925** | **0.8921** | 357.6 | 0.8874 | 0.8841 | 222.4 | **0.8893** | **0.8864** | 424.0 |
| | CD | 0.8293 | 0.8283 | 0.8282 | 497.0 | **0.8295** | **0.8294** | 349.8 | 0.8291 | 0.8286 | 375.2 | **0.8295** | 0.8276 | 304.0 |
| | GC | 0.7450 | 0.7630 | 0.7580 | 459.2 | **0.7640** | **0.7620** | 368.2 | 0.7510 | 0.7490 | 229.6 | **0.7680** | **0.7680** | 471.8 |
| Mean | | 0.6806 | 0.7326 | 0.7328 | 472.5 | **0.7408** | **0.7412** | 370.4 | 0.7285 | 0.7288 | 262.9 | **0.7392** | **0.7393** | 333.0 |
| Light-GBM | AF | 0.8375 | 0.8304 | 0.8356 | 479.6 | **0.8311** | **0.8392** | 380.2 | 0.8185 | 0.8266 | 284.6 | **0.8366** | **0.8395** | 360.6 |
| | BH | 0.5537 | 0.5503 | 0.5467 | 490.8 | **0.5619** | **0.5644** | 342.0 | 0.5500 | 0.5609 | 238.4 | **0.5642** | **0.5595** | 345.6 |
| | BS | 0.9429 | 0.9693 | 0.9691 | 480.2 | **0.9737** | **0.9754** | 380.0 | 0.9539 | 0.9536 | 312.6 | **0.9801** | **0.9813** | 236.8 |
| | WQR | 0.3825 | 0.4087 | 0.4151 | 493.0 | **0.4118** | **0.4171** | 322.8 | 0.4057 | 0.4050 | 246.8 | 0.4021 | 0.4042 | 293.6 |
| | ACT | 0.8832 | 0.8864 | 0.8883 | 513.0 | **0.8888** | **0.8925** | 367.4 | 0.8813 | 0.8748 | 229.0 | **0.8902** | **0.8925** | 359.6 |
| | CD | 0.8300 | 0.8284 | 0.8292 | 490.8 | **0.8301** | **0.8297** | 352.2 | 0.8295 | 0.8299 | 218.6 | **0.8303** | 0.8294 | 371.2 |
| | GC | 0.7250 | 0.7620 | 0.7620 | 482.4 | **0.7680** | **0.7720** | 376.6 | 0.7550 | 0.7550 | 225.0 | **0.7760** | **0.7700** | 382.2 |
| Mean | | 0.6806 | 0.7479 | 0.7494 | 490.0 | **0.7522** | **0.7558** | 360.2 | 0.7420 | 0.7437 | 250.7 | **0.7542** | **0.7538** | 335.7 |
| Mean | | 0.6806 | 0.7105 | 0.7118 | 469.9 | **0.7171** | **0.7185** | 362.7 | 0.7078 | 0.7092 | 255.9 | **0.7187** | **0.7183** | 331.9 |

tuning significantly outperforms all baselines other than DIFER at the $p = 0.05$ level. With GPT-4, the performance difference between PromptFE and CAAFE (Hollmann et al., 2023) as well as OCTree (Nam et al., 2024) is statistically significant at the $p = 0.05$ level.

**Table 10:** Full experimental results of Table 2 performance comparison of PromptFE with and without RPN canonicalization. For each compared version, the left and middle columns show the results without and with parameter tuning of the downstream model algorithm post AutoFE, respectively, and the right column shows the number of LLM generations. The results where the full version outperforms the reduced version are boldfaced.

| Model | Dataset | Raw | GPT-3.5 | | | | | | GPT-4 | | | | | |
| | | | w/o Canonicalization | | | PromptFE | | | w/o Canonicalization | | | PromptFE | | |
|---|---|---|---|---|---|---|---|---|---|---|---|---|---|---|
| Linear Model | AF | 0.3474 | 0.6679 | 0.6688 | 338.6 | 0.6612 | 0.6616 | 339.8 | 0.6538 | 0.6529 | 321.2 | **0.6649** | **0.6647** | 371.4 |
| | BH | 0.3776 | 0.5048 | 0.5076 | 351.2 | 0.4995 | 0.5025 | 378.6 | 0.4987 | 0.5030 | 310.8 | **0.5184** | **0.5289** | 335.4 |
| | BS | 1.0000 | – | – | – | – | – | – | – | – | – | – | – | – |
| | WQR | 0.2696 | 0.2702 | 0.2735 | 336.2 | **0.2722** | **0.2745** | **328.4** | 0.2690 | 0.2706 | 279.0 | **0.2713** | **0.2748** | 312.6 |
| | ACT | 0.8505 | 0.8748 | 0.8794 | 366.4 | 0.8729 | 0.8794 | 372.2 | 0.8738 | 0.8752 | 298.0 | **0.8766** | **0.8762** | 377.4 |
| | CD | 0.8267 | 0.8280 | 0.8290 | 350.4 | **0.8282** | 0.8282 | **342.0** | 0.8270 | 0.8271 | 285.4 | **0.8288** | **0.8288** | 250.4 |
| | GC | 0.7100 | 0.7370 | 0.7330 | 352.0 | **0.7570** | **0.7460** | 379.0 | 0.7550 | 0.7490 | 447.2 | **0.7590** | 0.7420 | 310.6 |
| Mean | | 0.5636 | 0.6471 | 0.6486 | 349.1 | **0.6485** | **0.6487** | 356.7 | 0.6462 | 0.6463 | 323.6 | **0.6532** | **0.6526** | 326.3 |
| Random Forests | AF | 0.7677 | 0.7628 | 0.7762 | 358.0 | **0.7709** | **0.7787** | 393.2 | 0.7743 | 0.7843 | 340.2 | 0.7681 | 0.7749 | **314.2** |
| | BH | 0.5426 | 0.5573 | 0.5573 | 364.0 | 0.5549 | 0.5522 | 374.4 | 0.5491 | 0.5460 | 322.4 | **0.5543** | **0.5522** | 278.6 |
| | BS | 0.9446 | 0.9804 | 0.9807 | 372.2 | **0.9873** | **0.9881** | 386.8 | 0.9778 | 0.9777 | 284.4 | **0.9845** | **0.9848** | 255.0 |
| | WQR | 0.3662 | 0.3776 | 0.3726 | 334.6 | **0.3862** | **0.3845** | 362.6 | 0.3739 | 0.3719 | 269.8 | **0.3810** | **0.3810** | 283.2 |
| | ACT | 0.8808 | 0.8879 | 0.8841 | 353.4 | **0.8925** | **0.8921** | 357.6 | 0.8841 | 0.8864 | 327.6 | **0.8893** | 0.8864 | 424.0 |
| | CD | 0.8293 | 0.8283 | 0.8285 | 381.6 | **0.8295** | **0.8294** | 349.8 | 0.8290 | 0.8287 | 297.2 | **0.8295** | 0.8276 | 304.0 |
| | GC | 0.7450 | 0.7660 | 0.7620 | 342.2 | 0.7640 | 0.7620 | 368.2 | 0.7680 | 0.7610 | 368.2 | 0.7680 | **0.7680** | 471.8 |
| Mean | | 0.6806 | 0.7372 | 0.7373 | 358.0 | **0.7408** | **0.7412** | 370.4 | 0.7366 | 0.7366 | 315.7 | **0.7392** | **0.7393** | 333.0 |
| Light-GBM | AF | 0.8375 | 0.8322 | 0.8365 | 343.6 | 0.8311 | **0.8392** | 380.2 | 0.8280 | 0.8350 | 376.0 | **0.8366** | **0.8395** | 360.6 |
| | BH | 0.5537 | 0.5599 | 0.5599 | 339.2 | **0.5619** | **0.5644** | 342.0 | 0.5577 | 0.5548 | 315.2 | **0.5642** | **0.5595** | 345.6 |
| | BS | 0.9429 | 0.9643 | 0.9664 | 368.8 | **0.9737** | **0.9754** | 380.0 | 0.9597 | 0.9609 | 276.2 | **0.9801** | **0.9813** | 236.8 |
| | WQR | 0.3825 | 0.4075 | 0.4042 | 346.4 | **0.4118** | **0.4171** | 322.8 | 0.4036 | 0.4032 | 288.2 | 0.4021 | **0.4042** | 293.6 |
| | ACT | 0.8832 | 0.8813 | 0.8860 | 342.4 | **0.8888** | **0.8925** | 367.4 | 0.8822 | 0.8879 | 313.2 | **0.8902** | **0.8925** | 359.6 |
| | CD | 0.8300 | 0.8302 | 0.8291 | 355.8 | 0.8301 | **0.8297** | 352.2 | 0.8295 | 0.8291 | 301.6 | **0.8303** | **0.8294** | 371.2 |
| | GC | 0.7250 | 0.7640 | 0.7650 | 346.2 | **0.7680** | **0.7720** | 376.6 | 0.7620 | 0.7650 | 428.8 | **0.7760** | **0.7700** | 382.2 |
| Mean | | 0.6806 | 0.7485 | 0.7490 | 348.9 | **0.7522** | **0.7558** | 360.2 | 0.7461 | 0.7480 | 328.5 | **0.7542** | 0.7538 | 335.7 |
| Mean | | 0.6806 | 0.7141 | 0.7148 | 352.2 | **0.7171** | **0.7185** | 362.7 | 0.7128 | 0.7135 | 322.5 | **0.7187** | 0.7183 | 331.9 |

**Table 11:** Standard deviations of Table 8 comparison of overall performance.

| Model | Dataset | Raw | DIFER | | OpenFE | | CAAFE | | | | OCTree | | PromptFE (ours) | | | |
| | | | | | | | GPT-3.5 | | GPT-4 | | | | GPT-3.5 | | GPT-4 | |
|---|---|---|---|---|---|---|---|---|---|---|---|---|---|---|---|---|
| Linear Model | AF | – | 0.2559 | 0.2012 | 0.0015 | 0.0014 | 0.0099 | 0.0102 | 0.0511 | 0.0513 | 0.0199 | 0.0199 | 0.0101 | 0.0100 | 0.0267 | 0.0268 |
| | BH | – | 0.0092 | 0.0153 | 0.0169 | 0.0188 | 0.0196 | 0.0184 | 0.0408 | 0.0419 | 0.0502 | 0.0516 | 0.0111 | 0.0149 | 0.0254 | 0.0184 |
| | WQR | – | 0.0305 | 0.0223 | 0.0058 | 0.0055 | 0.0046 | 0.0060 | 0.0060 | 0.0060 | 0.0045 | 0.0041 | 0.0135 | 0.0112 | 0.0068 | 0.0044 |
| | ACT | – | 0.0179 | 0.0073 | 0.0140 | 0.0105 | 0.0035 | 0.0021 | 0.0054 | 0.0053 | 0.0148 | 0.0148 | 0.0085 | 0.0051 | 0.0040 | 0.0062 |
| | CD | – | 0.0014 | 0.0006 | 0.0006 | 0.0002 | 0.0006 | 0.0007 | 0.0057 | 0.0051 | 0.0002 | 0.0003 | 0.0013 | 0.0007 | 0.0006 | 0.0009 |
| | GC | – | 0.0272 | 0.0104 | 0.0097 | 0.0076 | 0.0100 | 0.0125 | 0.0134 | 0.0108 | 0.0084 | 0.0079 | 0.0120 | 0.0213 | 0.0108 | 0.0152 |
| Random Forests | AF | – | 0.0054 | 0.0044 | 0.0032 | 0.0036 | 0.0032 | 0.0034 | 0.0108 | 0.0084 | 0.0084 | 0.0075 | 0.0090 | 0.0086 | 0.0059 | 0.0095 |
| | BH | – | 0.0142 | 0.0131 | 0.0034 | 0.0068 | 0.0050 | 0.0050 | 0.0084 | 0.0113 | 0.0052 | 0.0050 | 0.0057 | 0.0077 | 0.0059 | 0.0046 |
| | BS | – | 0.0128 | 0.0113 | 0.0003 | 0.0003 | 0.0003 | 0.0003 | 0.0208 | 0.0207 | 0.0016 | 0.0014 | 0.0088 | 0.0070 | 0.0157 | 0.0154 |
| | WQR | – | 0.0108 | 0.0109 | 0.0030 | 0.0076 | 0.0022 | 0.0022 | 0.0051 | 0.0051 | 0.0039 | 0.0040 | 0.0034 | 0.0069 | 0.0022 | 0.0026 |
| | ACT | – | 0.0048 | 0.0058 | 0.0037 | 0.0087 | 0.0030 | 0.0055 | 0.0020 | 0.0030 | 0.0063 | 0.0035 | 0.0055 | 0.0051 | 0.0043 | 0.0054 |
| | CD | – | 0.0010 | 0.0011 | 0.0003 | 0.0004 | 0.0005 | 0.0004 | 0.0008 | 0.0001 | 0.0009 | 0.0006 | 0.0011 | 0.0010 | 0.0009 | 0.0017 |
| | GC | – | 0.0184 | 0.0177 | 0.0154 | 0.0110 | 0.0082 | 0.0076 | 0.0065 | 0.0164 | 0.0160 | 0.0130 | 0.0114 | 0.0067 | 0.0097 | 0.0097 |
| Light-GBM | AF | – | 0.0029 | 0.0029 | 0.0058 | 0.0036 | 0.0067 | 0.0027 | 0.0072 | 0.0077 | 0.0107 | 0.0104 | 0.0129 | 0.0054 | 0.0061 | 0.0041 |
| | BH | – | 0.0147 | 0.0260 | 0.0128 | 0.0150 | 0.0114 | 0.0111 | 0.0145 | 0.0188 | 0.0127 | 0.0170 | 0.0169 | 0.0076 | 0.0134 | 0.0073 |
| | BS | – | 0.0092 | 0.0070 | 0.0007 | 0.0004 | 0.0159 | 0.0198 | 0.0056 | 0.0139 | 0.0162 | 0.0174 | 0.0151 | 0.0139 | 0.0033 | 0.0034 |
| | WQR | – | 0.0134 | 0.0164 | 0.0072 | 0.0133 | 0.0084 | 0.0080 | 0.0116 | 0.0134 | 0.0099 | 0.0113 | 0.0123 | 0.0085 | 0.0097 | 0.0092 |
| | ACT | – | 0.0048 | 0.0042 | 0.0068 | 0.0094 | 0.0061 | 0.0045 | 0.0045 | 0.0027 | 0.0017 | 0.0054 | 0.0027 | 0.0017 | 0.0050 | 0.0077 |
| | CD | – | 0.0009 | 0.0013 | 0.0004 | 0.0010 | 0.0008 | 0.0005 | 0.0010 | 0.0007 | 0.0005 | 0.0007 | 0.0004 | 0.0004 | 0.0004 | 0.0008 |
| | GC | – | 0.0141 | 0.0184 | 0.0184 | 0.0184 | 0.0222 | 0.0166 | 0.0079 | 0.0199 | 0.0146 | 0.0152 | 0.0076 | 0.0045 | 0.0096 | 0.0146 |

**Table 12:** Standard deviations of Table 9 performance comparison of PromptFE with and without dataset semantic context.

| Model | Dataset | Raw | GPT-3.5 w/o Semantic Context | | | PromptFE | | | GPT-4 w/o Semantic Context | | | PromptFE | | |
|---|---|---|---|---|---|---|---|---|---|---|---|---|---|---|
| Linear Model | AF | – | 0.0147 | 0.0156 | 36.1 | 0.0101 | 0.0100 | 28.8 | 0.0162 | 0.0161 | 25.8 | 0.0267 | 0.0268 | 92.3 |
| | BH | – | 0.0444 | 0.0519 | 39.0 | 0.0111 | 0.0149 | 42.2 | 0.0161 | 0.0131 | 66.7 | 0.0254 | 0.0184 | 58.6 |
| | WQR | – | 0.0133 | 0.0032 | 48.9 | 0.0135 | 0.0112 | 15.3 | 0.0128 | 0.0046 | 23.5 | 0.0068 | 0.0044 | 80.6 |
| | ACT | – | 0.0088 | 0.0107 | 15.4 | 0.0085 | 0.0051 | 17.5 | 0.0056 | 0.0085 | 15.5 | 0.0040 | 0.0062 | 54.8 |
| | CD | – | 0.0014 | 0.0003 | 27.6 | 0.0013 | 0.0007 | 13.1 | 0.0021 | 0.0011 | 13.2 | 0.0006 | 0.0009 | 14.8 |
| | GC | – | 0.0114 | 0.0042 | 32.3 | 0.0120 | 0.0213 | 14.3 | 0.0125 | 0.0114 | 11.0 | 0.0108 | 0.0152 | 36.4 |
| Random Forests | AF | – | 0.0086 | 0.0058 | 60.3 | 0.0090 | 0.0086 | 47.3 | 0.0092 | 0.0079 | 27.9 | 0.0059 | 0.0095 | 93.6 |
| | BH | – | 0.0068 | 0.0068 | 45.3 | 0.0057 | 0.0077 | 14.5 | 0.0142 | 0.0132 | 24.7 | 0.0059 | 0.0046 | 23.0 |
| | BS | – | 0.0186 | 0.0181 | 112.1 | 0.0088 | 0.0070 | 47.8 | 0.0103 | 0.0088 | 38.8 | 0.0157 | 0.0154 | 39.2 |
| | WQR | – | 0.0078 | 0.0081 | 40.5 | 0.0034 | 0.0069 | 18.5 | 0.0092 | 0.0075 | 19.1 | 0.0022 | 0.0026 | 45.2 |
| | ACT | – | 0.0099 | 0.0035 | 33.7 | 0.0055 | 0.0051 | 13.1 | 0.0100 | 0.0093 | 16.6 | 0.0043 | 0.0054 | 85.7 |
| | CD | – | 0.0015 | 0.0008 | 53.3 | 0.0011 | 0.0010 | 14.5 | 0.0005 | 0.0008 | 83.4 | 0.0009 | 0.0017 | 56.9 |
| | GC | – | 0.0067 | 0.0057 | 28.9 | 0.0114 | 0.0067 | 17.3 | 0.0210 | 0.0143 | 12.8 | 0.0097 | 0.0097 | 113.1 |
| Light-GBM | AF | – | 0.0104 | 0.0060 | 66.8 | 0.0129 | 0.0054 | 21.7 | 0.0142 | 0.0155 | 39.6 | 0.0061 | 0.0041 | 73.1 |
| | BH | – | 0.0131 | 0.0170 | 60.7 | 0.0169 | 0.0076 | 20.7 | 0.0119 | 0.0121 | 25.7 | 0.0134 | 0.0073 | 36.1 |
| | BS | – | 0.0152 | 0.0178 | 76.3 | 0.0151 | 0.0139 | 31.8 | 0.0048 | 0.0049 | 74.5 | 0.0033 | 0.0034 | 32.1 |
| | WQR | – | 0.0151 | 0.0028 | 36.9 | 0.0123 | 0.0085 | 17.3 | 0.0195 | 0.0190 | 21.1 | 0.0097 | 0.0092 | 46.3 |
| | ACT | – | 0.0021 | 0.0030 | 44.2 | 0.0027 | 0.0017 | 28.5 | 0.0042 | 0.0128 | 15.7 | 0.0050 | 0.0077 | 49.6 |
| | CD | – | 0.0011 | 0.0011 | 59.4 | 0.0004 | 0.0004 | 15.7 | 0.0007 | 0.0010 | 5.6 | 0.0004 | 0.0008 | 85.7 |
| | GC | – | 0.0130 | 0.0148 | 41.7 | 0.0076 | 0.0045 | 23.0 | 0.0117 | 0.0094 | 13.7 | 0.0096 | 0.0146 | 46.9 |

**Table 13:** Standard deviations of Table 10 performance comparison of PromptFE with and without RPN canonicalization.

| Model | Dataset | Raw | GPT-3.5 w/o Canonicalization | | | PromptFE | | | GPT-4 w/o Canonicalization | | | PromptFE | | |
|---|---|---|---|---|---|---|---|---|---|---|---|---|---|---|
| Linear Model | AF | – | 0.0117 | 0.0111 | 24.8 | 0.0101 | 0.0100 | 28.8 | 0.0112 | 0.0106 | 17.3 | 0.0267 | 0.0268 | 92.3 |
| | BH | – | 0.0081 | 0.0138 | 22.1 | 0.0111 | 0.0149 | 42.2 | 0.0249 | 0.0294 | 60.6 | 0.0254 | 0.0184 | 58.6 |
| | WQR | – | 0.0127 | 0.0083 | 34.7 | 0.0135 | 0.0112 | 15.3 | 0.0179 | 0.0070 | 42.0 | 0.0068 | 0.0044 | 80.6 |
| | ACT | – | 0.0069 | 0.0084 | 13.9 | 0.0085 | 0.0051 | 17.5 | 0.0074 | 0.0069 | 22.3 | 0.0040 | 0.0062 | 54.8 |
| | CD | – | 0.0008 | 0.0014 | 24.7 | 0.0013 | 0.0007 | 13.1 | 0.0016 | 0.0012 | 22.0 | 0.0006 | 0.0009 | 14.8 |
| | GC | – | 0.0246 | 0.0091 | 14.3 | 0.0120 | 0.0213 | 14.3 | 0.0132 | 0.0042 | 99.5 | 0.0108 | 0.0152 | 36.4 |
| Random Forests | AF | – | 0.0079 | 0.0121 | 27.7 | 0.0090 | 0.0086 | 47.3 | 0.0080 | 0.0036 | 32.9 | 0.0059 | 0.0095 | 93.6 |
| | BH | – | 0.0054 | 0.0054 | 31.1 | 0.0057 | 0.0077 | 14.5 | 0.0095 | 0.0055 | 56.1 | 0.0059 | 0.0046 | 23.0 |
| | BS | – | 0.0180 | 0.0174 | 11.9 | 0.0088 | 0.0070 | 47.8 | 0.0211 | 0.0210 | 25.6 | 0.0157 | 0.0154 | 39.2 |
| | WQR | – | 0.0036 | 0.0051 | 17.5 | 0.0034 | 0.0069 | 18.5 | 0.0081 | 0.0099 | 27.7 | 0.0022 | 0.0026 | 45.2 |
| | ACT | – | 0.0055 | 0.0094 | 15.2 | 0.0055 | 0.0051 | 13.1 | 0.0039 | 0.0056 | 29.2 | 0.0043 | 0.0054 | 85.7 |
| | CD | – | 0.0009 | 0.0007 | 21.1 | 0.0011 | 0.0010 | 14.5 | 0.0013 | 0.0008 | 36.0 | 0.0009 | 0.0017 | 56.9 |
| | GC | – | 0.0219 | 0.0182 | 27.2 | 0.0114 | 0.0067 | 17.3 | 0.0148 | 0.0055 | 19.7 | 0.0097 | 0.0097 | 113.1 |
| Light-GBM | AF | – | 0.0157 | 0.0102 | 21.4 | 0.0129 | 0.0054 | 21.7 | 0.0078 | 0.0066 | 62.1 | 0.0061 | 0.0041 | 73.1 |
| | BH | – | 0.0125 | 0.0096 | 18.3 | 0.0169 | 0.0076 | 20.7 | 0.0098 | 0.0090 | 32.4 | 0.0134 | 0.0073 | 36.1 |
| | BS | – | 0.0202 | 0.0190 | 21.5 | 0.0151 | 0.0139 | 31.8 | 0.0115 | 0.0115 | 49.9 | 0.0033 | 0.0034 | 32.1 |
| | WQR | – | 0.0083 | 0.0181 | 10.3 | 0.0123 | 0.0085 | 17.3 | 0.0100 | 0.0092 | 26.2 | 0.0097 | 0.0092 | 46.3 |
| | ACT | – | 0.0078 | 0.0065 | 12.6 | 0.0027 | 0.0017 | 28.5 | 0.0048 | 0.0029 | 21.3 | 0.0050 | 0.0077 | 49.6 |
| | CD | – | 0.0005 | 0.0011 | 14.9 | 0.0004 | 0.0004 | 15.7 | 0.0003 | 0.0005 | 22.3 | 0.0004 | 0.0008 | 85.7 |
| | GC | – | 0.0096 | 0.0184 | 14.7 | 0.0076 | 0.0045 | 23.0 | 0.0368 | 0.0194 | 126.6 | 0.0096 | 0.0146 | 46.9 |

**Table 14:** Standard deviations of Table 3 effect of the number of example features in the prompt.

| Model | Dataset | Number of Examples 1 | | 5 | | 10 | | 20 | |
|---|---|---|---|---|---|---|---|---|---|
| RF | AF | 0.0054 | 55.8 | 0.0035 | 45.0 | 0.0042 | 47.3 | 0.0056 | 24.0 |
| | WQR | 0.0088 | 19.6 | 0.0038 | 11.4 | 0.0027 | 18.5 | 0.0096 | 29.6 |
| | CD | 0.0005 | 46.5 | 0.0007 | 19.1 | 0.0004 | 14.5 | 0.0006 | 17.8 |
| LGBM | AF | 0.0065 | 103.2 | 0.0031 | 21.6 | 0.0044 | 21.7 | 0.0044 | 56.4 |
| | WQR | 0.0048 | 16.9 | 0.0057 | 32.4 | 0.0064 | 17.3 | 0.0064 | 26.5 |
| | CD | 0.0003 | 71.2 | 0.0002 | 39.0 | 0.0007 | 15.7 | 0.0005 | 17.5 |

**Table 15:** Standard deviations of Table 4 effect of temperature.

| Model | Dataset | Temperature 0.5 | | Temperature 1 | | Temperature 1.5 | |
|---|---|---|---|---|---|---|---|
| RF | AF | 0.0071 | 160.9 | 0.0042 | 47.3 | 0.0040 | 34.7 |
| | CD | 0.0005 | 324.3 | 0.0004 | 14.5 | 0.0005 | 64.1 |
| LGBM | AF | 0.0042 | 523.3 | 0.0044 | 21.7 | 0.0022 | 59.8 |
| | CD | 0.0008 | 174.7 | 0.0007 | 15.7 | 0.0005 | 73.0 |

**Table 16:** Percentage performance improvement of PromptFE over the baseline methods with GPT-3.5. For each compared method, the left and right columns show the results without and with parameter tuning of the downstream model algorithm post AutoFE, respectively.

| Model | Dataset | Raw | | DIFER | | OpenFE | | CAAFE | |
|---|---|---|---|---|---|---|---|---|---|
| Linear Model | AF | 90.34 | 90.46 | 12.65 | 8.64 | 53.77 | 53.77 | 64.86 | 64.76 |
| | BH | 32.27 | 33.06 | -0.37 | 0.61 | 28.06 | 29.51 | 4.32 | 5.46 |
| | WQR | 0.96 | 1.80 | 9.97 | 4.37 | 0.35 | 0.32 | -0.74 | -0.46 |
| | ACT | 2.64 | 3.41 | 0.16 | -0.05 | 0.00 | 0.75 | 2.47 | 3.29 |
| | CD | 0.18 | 0.19 | 0.10 | 0.03 | 0.21 | 0.17 | 0.20 | 0.19 |
| | GC | 6.62 | 5.07 | 6.02 | 0.54 | 3.42 | 2.47 | 2.99 | 1.77 |
| | Mean | 22.17 | 22.33 | 4.76 | 2.36 | 14.30 | 14.50 | 12.35 | 12.50 |
| Random Forests | AF | 0.42 | 1.44 | 0.78 | 0.02 | 1.72 | 1.37 | -0.02 | 1.23 |
| | BH | 2.26 | 1.97 | -2.95 | -2.95 | -1.92 | -1.55 | -0.13 | -0.41 |
| | BS | 4.52 | 4.60 | 0.08 | 0.10 | -0.29 | -0.21 | -0.43 | -0.35 |
| | WQR | 5.44 | 5.00 | 0.61 | 0.34 | 2.89 | 3.11 | 3.86 | 3.42 |
| | ACT | 1.33 | 1.27 | 0.32 | 0.26 | 1.06 | 0.90 | 1.11 | 0.74 |
| | CD | 0.02 | 0.01 | 0.12 | 0.04 | 0.09 | 0.10 | 0.05 | 0.06 |
| | GC | 2.55 | 2.28 | 1.19 | 1.60 | -0.13 | 0.66 | -0.65 | 0.00 |
| | Mean | 2.36 | 2.37 | 0.02 | -0.08 | 0.49 | 0.63 | 0.54 | 0.67 |
| Light-GBM | AF | -0.76 | 0.20 | 0.32 | -0.23 | 1.51 | 1.80 | -0.63 | 0.53 |
| | BH | 1.48 | 1.94 | 0.21 | 0.14 | -1.30 | 0.47 | 1.42 | 1.32 |
| | BS | 3.27 | 3.45 | -0.27 | -0.32 | -0.15 | -0.43 | 1.91 | 1.98 |
| | WQR | 7.67 | 9.04 | -0.63 | -0.27 | 5.66 | 7.40 | -0.29 | 3.36 |
| | ACT | 0.63 | 1.06 | 1.06 | 1.11 | 0.90 | 1.43 | 0.74 | 0.74 |
| | CD | 0.02 | -0.04 | 0.22 | 0.25 | 0.10 | 0.12 | 0.06 | -0.01 |
| | GC | 5.93 | 6.48 | 0.39 | 1.58 | 1.72 | 0.26 | 2.54 | 2.25 |
| | Mean | 2.60 | 3.16 | 0.18 | 0.32 | 1.21 | 1.58 | 0.82 | 1.45 |
| Mean | | 8.39 | 8.63 | 1.50 | 0.79 | 4.88 | 5.12 | 4.18 | 4.49 |

**Table 17:** Percentage performance improvement of PromptFE over the baseline methods with GPT-4. For each compared method, the left and right columns show the results without and with parameter tuning of the downstream model algorithm post AutoFE, respectively.

| Model | Dataset | Raw | | DIFER | | OpenFE | | CAAFE | | OCTree | |
|---|---|---|---|---|---|---|---|---|---|---|---|
| Linear Model | AF | 91.40 | 91.34 | 13.27 | 9.14 | 54.62 | 54.48 | 51.94 | 51.82 | 41.52 | 41.47 |
| | BH | 37.28 | 40.06 | 3.41 | 5.90 | 32.92 | 36.33 | 15.14 | 17.39 | 15.73 | 16.37 |
| | WQR | 0.64 | 1.94 | 9.62 | 4.51 | 0.03 | 0.46 | -2.25 | -0.99 | -2.18 | -1.07 |
| | ACT | 3.08 | 3.02 | 0.59 | -0.42 | 0.43 | 0.37 | 2.35 | 2.24 | 0.48 | 0.48 |
| | CD | 0.26 | 0.26 | 0.18 | 0.10 | 0.28 | 0.25 | 0.61 | 0.61 | 0.22 | 0.20 |
| | GC | 6.90 | 4.51 | 6.30 | 0.00 | 3.69 | 1.92 | 5.27 | 2.91 | 4.26 | 2.34 |
| | Mean | 23.26 | 23.52 | 5.56 | 3.21 | 15.33 | 15.64 | 12.17 | 12.33 | 10.00 | 9.97 |
| Random Forests | AF | 0.05 | 0.93 | 0.41 | -0.47 | 1.35 | 0.86 | -0.20 | 0.37 | 1.02 | 1.22 |
| | BH | 2.16 | 1.76 | -3.05 | -3.15 | -2.02 | -1.76 | 0.56 | 0.54 | 0.44 | 0.45 |
| | BS | 4.23 | 4.25 | -0.20 | -0.23 | -0.57 | -0.54 | 0.28 | 0.32 | -0.79 | -0.75 |
| | WQR | 4.03 | 4.03 | -0.74 | -0.58 | 1.51 | 2.16 | 3.17 | 3.17 | 4.25 | 4.22 |
| | ACT | 0.95 | 0.64 | -0.05 | -0.37 | 0.69 | 0.26 | 0.74 | 0.42 | 0.32 | 0.48 |
| | CD | 0.02 | -0.20 | 0.13 | -0.18 | 0.10 | -0.11 | 0.02 | -0.13 | 0.05 | -0.15 |
| | GC | 3.09 | 3.09 | 1.72 | 2.40 | 0.39 | 1.45 | 0.26 | 0.66 | 1.32 | 1.32 |
| | Mean | 2.08 | 2.07 | -0.26 | -0.37 | 0.21 | 0.33 | 0.69 | 0.76 | 0.94 | 0.97 |
| Light-GBM | AF | -0.11 | 0.24 | 0.98 | -0.19 | 2.18 | 1.84 | -0.75 | -0.36 | 1.61 | 2.51 |
| | BH | 1.90 | 1.04 | 0.63 | -0.74 | -0.89 | -0.41 | 3.00 | 1.70 | 3.68 | 2.88 |
| | BS | 3.94 | 4.08 | 0.38 | 0.28 | 0.51 | 0.17 | 3.72 | 3.44 | 1.14 | 0.84 |
| | WQR | 5.12 | 5.67 | -2.98 | -3.35 | 3.16 | 4.08 | 3.04 | 2.28 | 3.71 | 5.42 |
| | ACT | 0.79 | 1.06 | 1.22 | 1.11 | 1.06 | 1.43 | 0.85 | 1.22 | 0.26 | 1.17 |
| | CD | 0.04 | -0.07 | 0.24 | 0.21 | 0.12 | 0.08 | 0.03 | 0.00 | 0.12 | 0.02 |
| | GC | 7.03 | 6.21 | 1.44 | 1.32 | 2.78 | 0.00 | 4.16 | -0.26 | 4.16 | 2.53 |
| | Mean | 2.67 | 2.60 | 0.27 | -0.19 | 1.27 | 1.03 | 2.01 | 1.15 | 2.10 | 2.20 |
| Mean | | 8.64 | 8.69 | 1.67 | 0.76 | 5.12 | 5.17 | 4.60 | 4.37 | 4.06 | 4.10 |

**Table 18:** The Nemenyi post-hoc test $p$-values for pairwise comparison of the methods in Table 1. Results that are significant at the $p = 0.05$ confidence level are boldfaced.

| | | Raw | DIFER | | OpenFE | | CAAFE GPT-3.5 | | CAAFE GPT-4 | | OCTree GPT-4 | | PromptFE GPT-3.5 | | PromptFE GPT-4 | |
|---|---|---|---|---|---|---|---|---|---|---|---|---|---|---|---|---|
| Raw | | 1.0000 | **0.0010** | **0.0010** | **0.0215** | **0.0117** | **0.0010** | **0.0010** | **0.0298** | **0.0140** | **0.0010** | **0.0114** | **0.0010** | **0.0010** | **0.0010** | **0.0010** |
| DIFER | | **0.0010** | 1.0000 | 0.4574 | 0.3602 | 0.4794 | 0.9000 | 0.9000 | 0.2980 | 0.4462 | 0.8973 | 0.4847 | 0.0674 | **0.0010** | **0.0066** | 0.2267 |
| | | **0.0010** | 0.4574 | 1.0000 | **0.0010** | **0.0010** | 0.0513 | **0.0500** | **0.0010** | **0.0010** | **0.0017** | **0.0010** | 0.9000 | 0.7955 | 0.9000 | 0.9000 |
| OpenFE | | **0.0215** | 0.3602 | **0.0010** | 1.0000 | 0.9000 | 0.9000 | 0.9000 | 0.9000 | 0.9000 | 0.9000 | 0.9000 | **0.0010** | **0.0010** | **0.0010** | **0.0010** |
| | | **0.0117** | 0.4794 | **0.0010** | 0.9000 | 1.0000 | 0.9000 | 0.9000 | 0.9000 | 0.9000 | 0.9000 | 0.9000 | **0.0010** | **0.0010** | **0.0010** | **0.0010** |
| CAAFE | GPT-3.5 | **0.0010** | 0.9000 | 0.0513 | 0.9000 | 0.9000 | 1.0000 | 0.9000 | 0.8922 | 0.9000 | 0.9000 | 0.9000 | **0.0025** | **0.0010** | **0.0010** | **0.0153** |
| | | **0.0010** | 0.9000 | **0.0500** | 0.9000 | 0.9000 | 0.9000 | 1.0000 | 0.8973 | 0.9000 | 0.9000 | 0.9000 | **0.0024** | **0.0010** | **0.0010** | **0.0148** |
| | GPT-4 | **0.0298** | 0.2980 | **0.0010** | 0.9000 | 0.9000 | 0.8922 | 0.8973 | 1.0000 | 0.9000 | 0.9000 | 0.9000 | **0.0010** | **0.0010** | **0.0010** | **0.0010** |
| | | **0.0140** | 0.4462 | **0.0010** | 0.9000 | 0.9000 | 0.9000 | 0.9000 | 0.9000 | 1.0000 | 0.9000 | 0.9000 | **0.0010** | **0.0010** | **0.0010** | **0.0010** |
| OCTree | GPT-4 | **0.0010** | 0.8973 | **0.0017** | 0.9000 | 0.9000 | 0.9000 | 0.9000 | 0.9000 | 0.9000 | 1.0000 | 0.9000 | **0.0010** | **0.0010** | **0.0010** | **0.0010** |
| | | **0.0114** | 0.4847 | **0.0010** | 0.9000 | 0.9000 | 0.9000 | 0.9000 | 0.9000 | 0.9000 | 0.9000 | 1.0000 | **0.0010** | **0.0010** | **0.0010** | **0.0010** |
| PromptFE | GPT-3.5 | **0.0010** | 0.0674 | 0.9000 | **0.0010** | **0.0010** | **0.0025** | **0.0024** | **0.0010** | **0.0010** | **0.0010** | **0.0010** | 1.0000 | 0.9000 | 0.9000 | 0.9000 |
| | | **0.0010** | **0.0010** | 0.7955 | **0.0010** | **0.0010** | **0.0010** | **0.0010** | **0.0010** | **0.0010** | **0.0010** | **0.0010** | 0.9000 | 1.0000 | 0.9000 | 0.9000 |
| | GPT-4 | **0.0010** | **0.0066** | 0.9000 | **0.0010** | **0.0010** | **0.0010** | **0.0010** | **0.0010** | **0.0010** | **0.0010** | **0.0010** | 0.9000 | 0.9000 | 1.0000 | 0.9000 |
| | | **0.0010** | 0.2267 | 0.9000 | **0.0010** | **0.0010** | **0.0153** | **0.0148** | **0.0010** | **0.0010** | **0.0010** | **0.0010** | 0.9000 | 0.9000 | 0.9000 | 1.0000 |

**Table 19:** The Nemenyi post-hoc test $p$-values for pairwise comparison of the methods in Table 1 excluding linear model results. Results that are significant at the $p = 0.05$ confidence level are boldfaced.

| | | Raw | DIFER | | OpenFE | | CAAFE GPT-3.5 | | CAAFE GPT-4 | | OCTree GPT-4 | | PromptFE GPT-3.5 | | PromptFE GPT-4 | |
|---|---|---|---|---|---|---|---|---|---|---|---|---|---|---|---|---|
| Raw | | 1.0000 | **0.0011** | **0.0010** | 0.5553 | 0.4753 | **0.0010** | **0.0014** | 0.4488 | 0.3036 | 0.6222 | 0.9000 | **0.0010** | **0.0010** | **0.0010** | **0.0010** |
| DIFER | | **0.0011** | 1.0000 | 0.9000 | 0.7316 | 0.8107 | 0.9000 | 0.9000 | 0.8350 | 0.9000 | 0.6648 | 0.1552 | 0.5614 | **0.0404** | 0.2187 | 0.9000 |
| | | **0.0010** | 0.9000 | 1.0000 | **0.0235** | **0.0357** | 0.9000 | 0.9000 | **0.0404** | 0.0775 | **0.0162** | **0.0010** | 0.9000 | 0.8350 | 0.9000 | 0.9000 |
| OpenFE | | 0.5553 | 0.7316 | **0.0235** | 1.0000 | 0.9000 | 0.6830 | 0.7681 | 0.9000 | 0.9000 | 0.9000 | 0.9000 | **0.0011** | **0.0010** | **0.0010** | **0.0250** |
| | | 0.4753 | 0.8107 | **0.0357** | 0.9000 | 1.0000 | 0.7620 | 0.8472 | 0.9000 | 0.9000 | 0.9000 | 0.9000 | **0.0019** | **0.0010** | **0.0010** | **0.0380** |
| CAAFE | GPT-3.5 | **0.0010** | 0.9000 | 0.9000 | 0.6830 | 0.7620 | 1.0000 | 0.9000 | 0.7864 | 0.9000 | 0.6161 | 0.1267 | 0.6100 | 0.0514 | 0.2588 | 0.9000 |
| | | **0.0014** | 0.9000 | 0.9000 | 0.7681 | 0.8472 | 0.9000 | 1.0000 | 0.8715 | 0.9000 | 0.7012 | 0.1783 | 0.5249 | **0.0335** | 0.1912 | 0.9000 |
| | GPT-4 | 0.4488 | 0.8350 | **0.0404** | 0.9000 | 0.9000 | 0.7864 | 0.8715 | 1.0000 | 0.9000 | 0.9000 | 0.9000 | **0.0022** | **0.0010** | **0.0010** | **0.0430** |
| | | 0.3036 | 0.9000 | 0.0775 | 0.9000 | 0.9000 | 0.9000 | 0.9000 | 0.9000 | 1.0000 | 0.9000 | 0.9000 | **0.0053** | **0.0010** | **0.0010** | 0.0819 |
| OCTree | GPT-4 | 0.6222 | 0.6648 | **0.0162** | 0.9000 | 0.9000 | 0.6161 | 0.7012 | 0.9000 | 0.9000 | 1.0000 | 0.9000 | **0.0010** | **0.0010** | **0.0010** | **0.0174** |
| | | 0.9000 | 0.1552 | **0.0010** | 0.9000 | 0.9000 | 0.1267 | 0.1783 | 0.9000 | 0.9000 | 0.9000 | 1.0000 | **0.0010** | **0.0010** | **0.0010** | **0.0010** |
| PromptFE | GPT-3.5 | **0.0010** | 0.5614 | 0.9000 | **0.0011** | **0.0019** | 0.6100 | 0.5249 | **0.0022** | **0.0053** | **0.0010** | **0.0010** | 1.0000 | 0.9000 | 0.9000 | 0.9000 |
| | | **0.0010** | **0.0404** | 0.8350 | **0.0010** | **0.0010** | 0.0514 | **0.0335** | **0.0010** | **0.0010** | **0.0010** | **0.0010** | 0.9000 | 1.0000 | 0.9000 | 0.8229 |
| | GPT-4 | **0.0010** | 0.2187 | 0.9000 | **0.0010** | **0.0010** | 0.2588 | 0.1912 | **0.0010** | **0.0010** | **0.0010** | **0.0010** | 0.9000 | 0.9000 | 1.0000 | 0.9000 |
| | | **0.0010** | 0.9000 | 0.9000 | **0.0250** | **0.0380** | 0.9000 | 0.9000 | **0.0430** | 0.0819 | **0.0174** | **0.0010** | 0.9000 | 0.8229 | 0.9000 | 1.0000 |

**Table 20:** The Nemenyi post-hoc test $p$-values for pairwise comparison of the methods in Table 2 performance comparison of PromptFE with and without dataset semantic context. Results that are significant at the $p = 0.05$ confidence level are boldfaced.

| | | Raw | GPT-3.5 Blinded | | GPT-3.5 PromptFE | | GPT-4 Blinded | | GPT-4 PromptFE | |
|---|---|---|---|---|---|---|---|---|---|---|
| Raw | | 1.0000 | **0.0010** | **0.0010** | **0.0010** | **0.0010** | **0.0017** | **0.0010** | **0.0010** | **0.0010** |
| GPT-3.5 | Blinded | **0.0010** | 1.0000 | 0.9000 | **0.0062** | **0.0010** | 0.9000 | 0.9000 | **0.0010** | **0.0057** |
| | | **0.0010** | 0.9000 | 1.0000 | 0.1775 | **0.0066** | 0.3858 | 0.9000 | **0.0105** | 0.1677 |
| | PromptFE | **0.0010** | **0.0062** | 0.1775 | 1.0000 | 0.9000 | **0.0010** | **0.0069** | 0.9000 | 0.9000 |
| | | **0.0010** | **0.0010** | **0.0066** | 0.9000 | 1.0000 | **0.0010** | **0.0010** | 0.9000 | 0.9000 |
| GPT-4 | Blinded | **0.0017** | 0.9000 | 0.3858 | **0.0010** | **0.0010** | 1.0000 | 0.9000 | **0.0010** | **0.0010** |
| | | **0.0010** | 0.9000 | 0.9000 | **0.0069** | **0.0010** | 0.9000 | 1.0000 | **0.0010** | **0.0062** |
| | PromptFE | **0.0010** | **0.0010** | **0.0105** | 0.9000 | 0.9000 | **0.0010** | **0.0010** | 1.0000 | 0.9000 |
| | | **0.0010** | **0.0057** | 0.1677 | 0.9000 | 0.9000 | **0.0010** | **0.0062** | 0.9000 | 1.0000 |

**Table 21:** The Nemenyi post-hoc test $p$-values for pairwise comparison of the methods in Table 2 performance comparison of PromptFE with and without RPN canonicalization. Results that are significant at the $p = 0.05$ confidence level are boldfaced.

| | | Raw | GPT-3.5 w/o | | GPT-3.5 PromptFE | | GPT-4 w/o | | GPT-4 PromptFE | |
|---|---|---|---|---|---|---|---|---|---|---|
| Raw | | 1.0000 | **0.0010** | **0.0010** | **0.0010** | **0.0010** | **0.0010** | **0.0010** | **0.0010** | **0.0010** |
| GPT-3.5 | w/o | **0.0010** | 1.0000 | 0.9000 | 0.2977 | **0.0060** | 0.9000 | 0.9000 | **0.0224** | 0.4293 |
| | | **0.0010** | 0.9000 | 1.0000 | 0.6618 | **0.0433** | 0.8811 | 0.8889 | 0.1230 | 0.7871 |
| | PromptFE | **0.0010** | 0.2977 | 0.6618 | 1.0000 | 0.9000 | **0.0341** | **0.0355** | 0.9000 | 0.9000 |
| | | **0.0010** | **0.0060** | **0.0433** | 0.9000 | 1.0000 | **0.0010** | **0.0010** | 0.9000 | 0.8028 |
| GPT-4 | w/o | **0.0010** | 0.9000 | 0.8811 | **0.0341** | **0.0010** | 1.0000 | 0.9000 | **0.0010** | 0.0635 |
| | | **0.0010** | 0.9000 | 0.8889 | **0.0355** | **0.0010** | 0.9000 | 1.0000 | **0.0010** | 0.0659 |
| | PromptFE | **0.0010** | **0.0224** | 0.1230 | 0.9000 | 0.9000 | **0.0010** | **0.0010** | 1.0000 | 0.9000 |
| | | **0.0010** | 0.4293 | 0.7871 | 0.9000 | 0.8028 | 0.0635 | 0.0659 | 0.9000 | 1.0000 |

## D.8 Additional Hyperparameter Effect

**Number of Iterations.** Figure 13 shows the validation scores on the *AF* and *CD* datasets, which contain the smallest and largest numbers of features, respectively, using Random Forests and Light-GBM. The validation score is evaluated after adding the selected set of candidate features to the dataset, as denoted by $s_{n*}$ in line 17 of Algorithm 1. We terminate our algorithm once we have 200 candidate features, as constructing additional features does not substantially enhance the performance, but constructing fewer features degrades the performance in some cases.

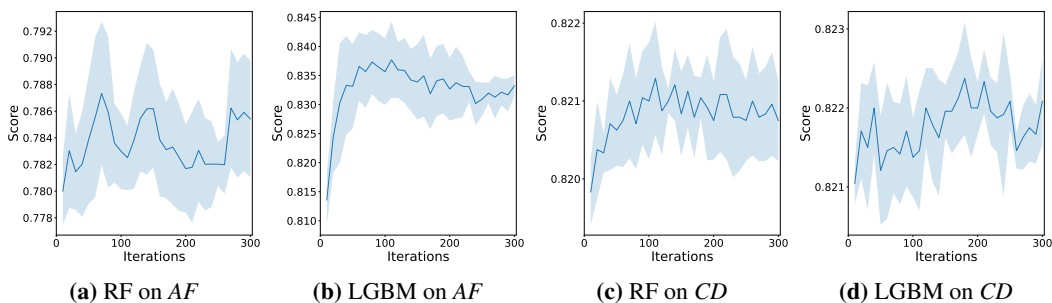

| (a) RF on *AF* | (b) LGBM on *AF* | (c) RF on *CD* | (d) LGBM on *CD* |

**Figure 13:** The validation score across iterations using Random Forests and LightGBM.

**Number of Examples in Prompt with GPT-4.** Table 22 reports the maximum validation score across iterations by varying the number of example features provided in the prompt to GPT-4. We observe improved performance as the number of example features increases. This suggests that providing more example features helps fully leverage GPT-4's enhanced in-context learning capabilities. In our experiments, we set the number of example features to 10 for a fair comparison with GPT-3.5.

**Table 22:** Effect of the number of example features in the prompt with GPT-4. For each compared setting, the left column shows the validation score, and the right column shows the number of LLM generations. The best results are boldfaced.

| Model | Dataset | Number of Examples | | | | |
|-------|---------|--------|--------|--------|--------|--------|
| | | 1 | 5 | 10 | 20 | 30 |
| RF | AF | 0.7864 | **0.7922** | 0.7905 | 0.7897 | 0.7920 |
| | WQR | 0.3847 | 0.3835 | 0.3839 | 0.3862 | **0.3862** |
| | CD | 0.8219 | 0.8218 | 0.8218 | 0.8219 | **0.8222** |
| LGBM | AF | 0.8387 | 0.8413 | 0.8401 | **0.8433** | 0.8411 |
| | WQR | 0.4216 | 0.4242 | **0.4290** | 0.4258 | 0.4267 |
| | CD | 0.8231 | **0.8234** | 0.8227 | 0.8229 | 0.8231 |
| Mean | | 0.6794 | 0.6810 | 0.6813 | 0.6816 | **0.6819** |

## D.9 NUMBER OF SELECTED FEATURES

Table 23 compares the number of features added to the datasets. Our method PromptFE adaptively determines the number of features and selects fewer features than DIFER (Zhu et al., 2022b), demonstrating the effectiveness of the features generated by our method.

**Table 23:** Comparison of the number of selected features.

| Model | Dataset | DIFER | OpenFE | PromptFE Blinded | | PromptFE | |
|---|---|---|---|---|---|---|---|
| | | | | GPT-3.5 | GPT-4 | GPT-3.5 | GPT-4 |
| Linear Model | AF | 310 | 10 | 167 | 165 | 162 | 183 |
| | BH | 156 | 10 | 104 | 141 | 144 | 90 |
| | WQR | 109 | 10 | 57 | 80 | 43 | 55 |
| | ACT | 113 | 10 | 84 | 49 | 85 | 14 |
| | CD | 157 | 10 | 92 | 68 | 74 | 74 |
| | GC | 105 | 10 | 75 | 97 | 120 | 51 |
| Random Forests | AF | 387 | 10 | 39 | 19 | 15 | 34 |
| | BH | 186 | 10 | 4 | 6 | 19 | 77 |
| | BS | 46 | 10 | 9 | 7 | 9 | 65 |
| | WQR | 63 | 10 | 9 | 44 | 39 | 45 |
| | ACT | 339 | 10 | 55 | 35 | 69 | 61 |
| | CD | 178 | 10 | 97 | 74 | 94 | 89 |
| | GC | 92 | 10 | 68 | 84 | 31 | 59 |
| Light-GBM | AF | 325 | 10 | 30 | 55 | 42 | 24 |
| | BH | 118 | 10 | 15 | 17 | 16 | 25 |
| | BS | 287 | 10 | 119 | 48 | 68 | 116 |
| | WQR | 454 | 10 | 64 | 29 | 129 | 128 |
| | ACT | 132 | 10 | 54 | 46 | 16 | 51 |
| | CD | 409 | 10 | 68 | 53 | 12 | 50 |
| | GC | 501 | 10 | 61 | 86 | 16 | 35 |
| Mean | | 223 | 10 | 64 | 60 | 60 | 66 |

## D.10 COMPUTATION COST

Table 24 compares the number of features evaluated during the feature search process. Guided by domain knowledge, our method PromptFE evaluates much fewer features than DIFER (Zhu et al., 2022b) and OpenFE (Zhang et al., 2023).

Tables 25 and 26 summarize the computation time, with *gpt-3.5-turbo-0125* as the LLM. For PromptFE, the computation time of LLM generation and feature evaluation is relatively stable across datasets of varying sizes. We note that the LLM generation time can be substantially reduced by instructing the LLM to generate multiple features in a generation step.

The sizes of datasets were listed in Table 5. We see that only the feature selection time is sensitive to dataset sizes. While in general the cost of downstream model evaluations grows proportionally with the dataset size, the actual cost depends on the hyperparameter of the downstream model, e.g., the maximum tree depth. Based on the observations in Figure 10, the LLM generation time is roughly constant across iterations. The feature evaluation time is also constant. The feature selection time scales quadratically with the number of candidate features but can be computed in parallel.

**Table 24:** Comparison of the number of evaluated features during feature search.

| Model | Dataset | DIFER | OpenFE | PromptFE |
|---|---|---|---|---|
| Linear Model | AF | 2083 | 224 | 200 |
| | BH | 2081 | 1167 | 200 |
| | WQR | 2083 | 929 | 200 |
| | ACT | 2077 | 4310 | 200 |
| | CD | 2088 | 3385 | 200 |
| | GC | 2076 | 4169 | 200 |
| Random Forests | AF | 2085 | 224 | 200 |
| | BH | 2079 | 1051 | 200 |
| | BS | 2082 | 310 | 200 |
| | WQR | 2085 | 929 | 200 |
| | ACT | 2079 | 1636 | 200 |
| | CD | 2086 | 1801 | 200 |
| | GC | 2078 | 2139 | 200 |
| Light-GBM | AF | 2084 | 224 | 200 |
| | BH | 2080 | 1051 | 200 |
| | BS | 2083 | 310 | 200 |
| | WQR | 2084 | 929 | 200 |
| | ACT | 2079 | 1636 | 200 |
| | CD | 2087 | 1801 | 200 |
| | GC | 2078 | 2139 | 200 |
| Mean | | 2082 | 1518 | 200 |

## E EXPERIMENTS ON PROPRIETARY DATASETS

We have conducted experiments on our proprietary real-world dataset containing over 100,000 samples and over 1,000 features, where most features contain a substantial proportion of missing values. We select features of top 100 mutual information scores with the target, which filters out features with too many missing values, and perform AutoFE on those features. With preprocessing, PromptFE brings significant performance improvements to downstream models on our proprietary real-world dataset.

Table 25: Comparison of computation time, in minutes.

| Model | Dataset | DIFER | OpenFE | CAAFE | PromptFE |
|---|---|---|---|---|---|
| Linear Model | AF | 33.49 | 0.21 | 1.73 | 42.80 |
| | BH | 41.17 | 0.21 | 1.18 | 41.28 |
| | WQR | 34.94 | 0.25 | 1.21 | 42.33 |
| | ACT | 44.18 | 0.40 | 1.25 | 43.60 |
| | CD | 433.94 | 1.49 | 3.17 | 57.82 |
| | GC | 29.30 | 0.37 | 1.68 | 43.71 |
| Random Forests | AF | 178.50 | 0.23 | 4.22 | 63.30 |
| | BH | 89.07 | 0.24 | 5.52 | 51.70 |
| | BS | 98.50 | 0.23 | 4.05 | 51.13 |
| | WQR | 298.46 | 0.29 | 9.35 | 63.12 |
| | ACT | 78.44 | 0.28 | 3.82 | 44.66 |
| | CD | 571.33 | 1.12 | 14.05 | 94.08 |
| | GC | 60.41 | 0.28 | 3.24 | 45.06 |
| Light-GBM | AF | 301.56 | 0.25 | 5.81 | 63.06 |
| | BH | 62.30 | 0.24 | 3.01 | 44.84 |
| | BS | 74.59 | 0.24 | 2.55 | 45.23 |
| | WQR | 361.19 | 0.29 | 5.68 | 58.97 |
| | ACT | 36.39 | 0.28 | 1.73 | 42.71 |
| | CD | 102.04 | 1.07 | 2.49 | 46.34 |
| | GC | 48.63 | 0.28 | 2.97 | 43.03 |
| Mean | | 148.92 | 0.41 | 3.94 | 51.44 |

Table 26: Computation time of different components of PromptFE, in minutes.

| Model | Dataset | LLM Generation | Feature Evaluation | Feature Selection |
|---|---|---|---|---|
| Linear Model | AF | 16.73 | 22.98 | 3.08 |
| | BH | 18.50 | 20.18 | 2.60 |
| | WQR | 19.07 | 20.24 | 3.02 |
| | ACT | 18.92 | 20.97 | 3.71 |
| | CD | 16.73 | 25.14 | 15.95 |
| | GC | 17.01 | 23.24 | 3.47 |
| Random Forests | AF | 15.34 | 25.32 | 22.64 |
| | BH | 18.60 | 23.69 | 9.41 |
| | BS | 15.12 | 25.16 | 10.87 |
| | WQR | 12.75 | 23.81 | 26.56 |
| | ACT | 13.79 | 21.67 | 9.20 |
| | CD | 12.48 | 25.89 | 55.71 |
| | GC | 14.80 | 21.91 | 8.35 |
| Light-GBM | AF | 17.37 | 21.06 | 24.63 |
| | BH | 19.70 | 20.40 | 4.74 |
| | BS | 17.03 | 22.18 | 6.02 |
| | WQR | 16.27 | 21.19 | 21.51 |
| | ACT | 19.18 | 20.24 | 3.29 |
| | CD | 16.53 | 21.68 | 8.13 |
| | GC | 17.00 | 20.40 | 5.63 |
| Mean | | 16.65 | 22.37 | 12.43 |

# F ADDITIONAL ANALYSIS

## F.1 FEATURE ANALYSIS

Figure 14 compares the proportions of generated features selecting each feature attribute across different datasets and downstream models (linear models and Random Forests) for both the full and semantically blinded versions of PromptFE. In the blinded version, we observe that the LLM tends to prioritize earlier feature attributes in the dataset while paying less attention to later ones, reflecting an inherent bias of the language model. In contrast, in the full version, the selection of feature attributes is guided by the semantic information of the dataset rather than the positional order of the attributes. Specifically, Attribute 19 *CD4 at baseline* in AIDS Clinical Trials (ACT) and Attribute 10 *alcohol* in Wine Quality Red (WQR), which contain useful information for predicting the targets *censoring indicator* and *quality*, respectively, are included in the majority of the generated features. This demonstrates the role of dataset semantic information in the LLM-based feature search process.

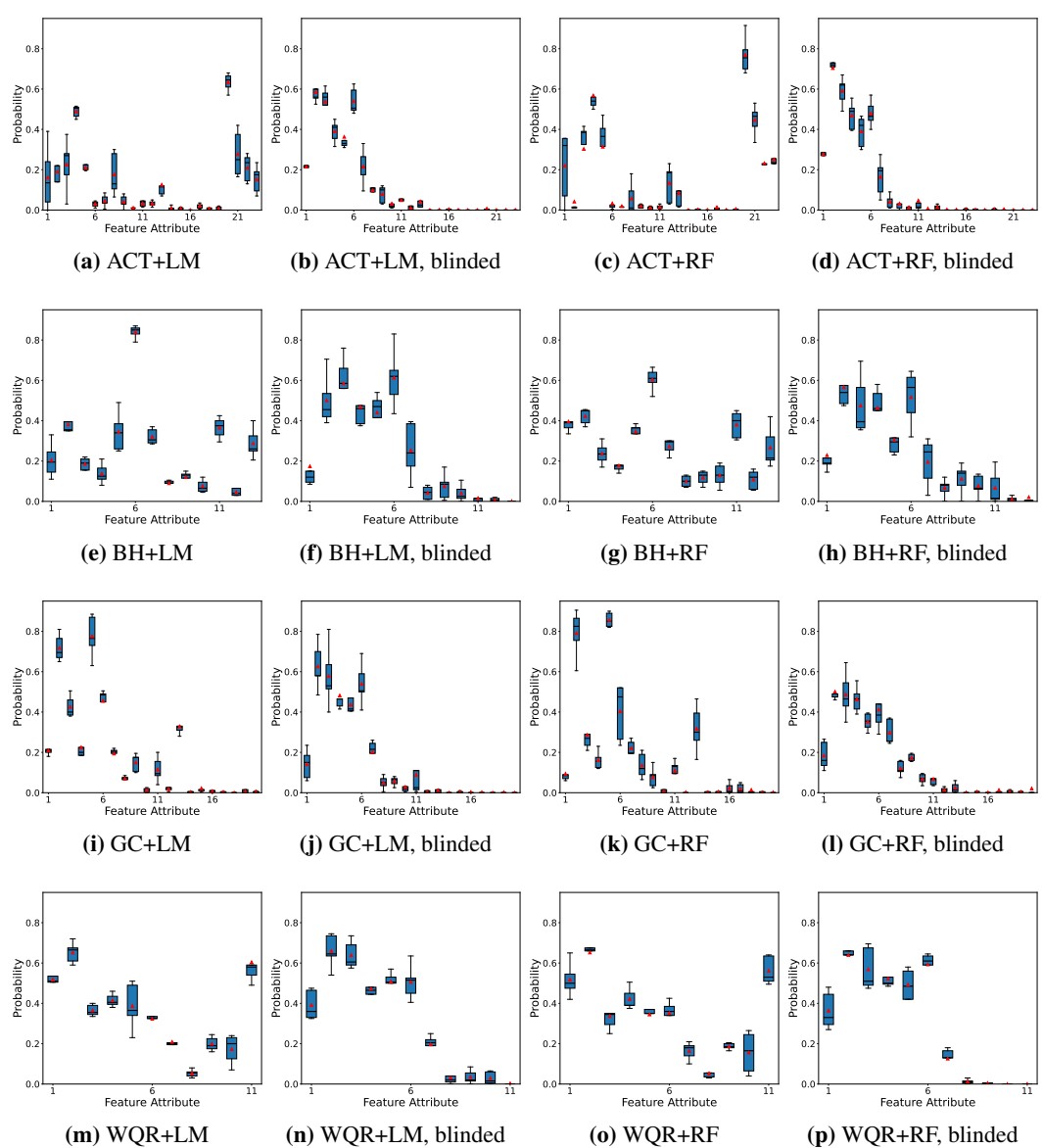

**(a)** ACT+LM     **(b)** ACT+LM, blinded     **(c)** ACT+RF     **(d)** ACT+RF, blinded

**(e)** BH+LM     **(f)** BH+LM, blinded     **(g)** BH+RF     **(h)** BH+RF, blinded

**(i)** GC+LM     **(j)** GC+LM, blinded     **(k)** GC+RF     **(l)** GC+RF, blinded

**(m)** WQR+LM     **(n)** WQR+LM, blinded     **(o)** WQR+RF     **(p)** WQR+RF, blinded

**Figure 14:** Distributions of feature attribute selection in the constructed features with GPT-4.

## F.2 FEATURE IMPORTANCE

Figure 15 shows the feature importance scores across different datasets and downstream models. We employ magnitudes of coefficients for linear models, impurity-based feature importance for Random Forests (Breiman, 2001), and total gains of splits for LightGBM (Ke et al., 2017). PromptFE augments datasets with the constructed features extracting important information for target prediction. We observe that Random Forests and LightGBM benefit from features of higher orders compared to linear models, since they are capable of synthesizing simple features internally. Our approach adapts the feature complexity for different downstream models.

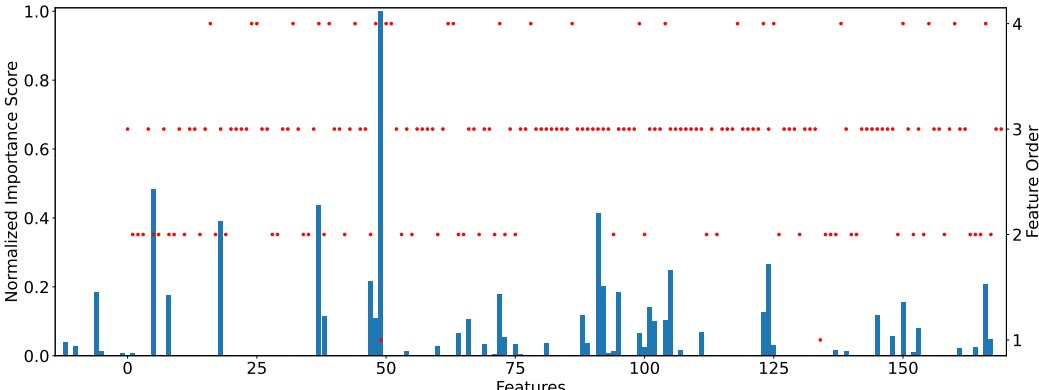

**(a)** BH+LM. Test performance improves from to 0.3776 to 0.5157.

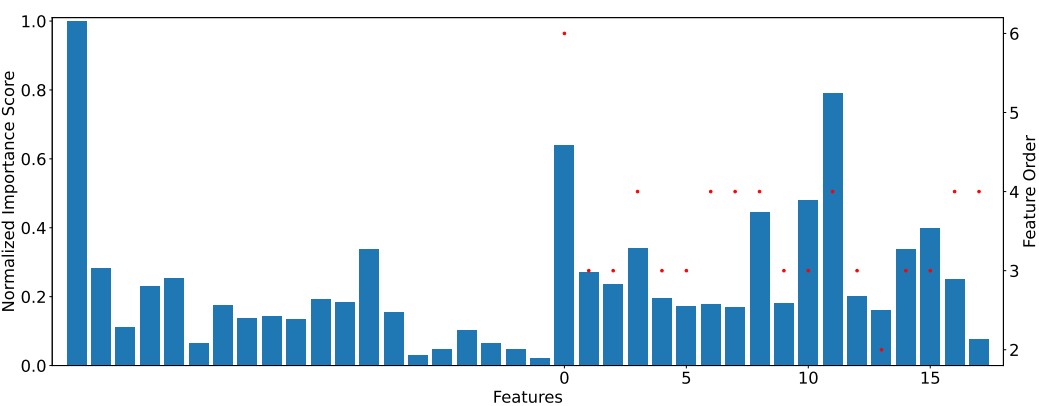

**(b)** GC+RF. Test performance improves from to 0.7450 to 0.7700.

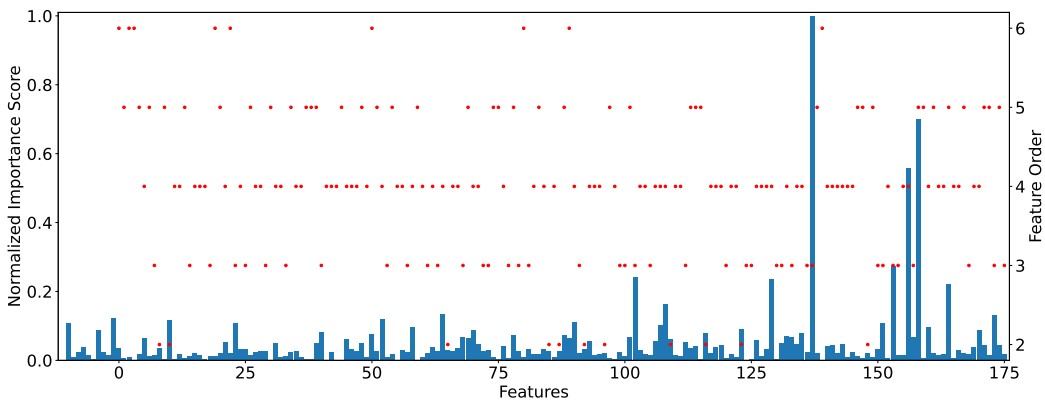

**(c)** WQR+LGBM. Test performance improves from to 0.3825 to 0.4299.

**Figure 15:** The blue bars show the normalized feature importance scores. The red dots show the order of features. The constructed features positioned on the x-axis start at index 0, following raw dataset features.

### F.3 FEATURE SEARCH EXAMPLE

Figure 16 shows samples of the LLM-based feature search process from a single run. In iteration 155, the generated feature is derived from the combination of three example features. In iteration 158, the generated feature is derived from the combination of two example features, with the feature attributes modified. Iteration 160 exhibits an instance of modification on the example feature. Iteration 161 exhibits an instance of analogy of the example feature.

Figure 16: Examples of feature search. The prompt examples that relate to the generated feature are highlighted in red.

Iteration 155
Generated feature:
6, 12, division, 18, *, min_max
Prompt examples:
4, log, 13, *, 17, min_max, sqrt_abs, +
6, 12, *, 21, sqrt_abs, +
4, 14, +, 18, *, min_max
11, 21, +, 6, *, sqrt_abs
2, 4, +, 6, division, log
4, 6, +, 16, *, 17, +, min_max
6, 21, division, 17, log, *
12, min_max, 18, *, 5, mod_column
2, 6, mod_column, 17, +, sqrt_abs
2, 3, +, 4, sqrt_abs, *
4, 12, division, 16, *, min_max, sqrt_abs

Iteration 158
Generated feature:
5, log, 19, *, 15, +, min_max
Prompt examples:
4, log, 13, *, 17, min_max, sqrt_abs, +
6, 12, *, 21, sqrt_abs, +
4, 14, +, 18, *, min_max
11, 21, +, 6, *, sqrt_abs
2, 4, +, 6, division, log
4, 6, +, 16, *, 17, +, min_max
6, 21, division, 17, log, *
12, min_max, 18, *, 5, mod_column
2, 6, mod_column, 17, +, sqrt_abs
2, 3, +, 4, sqrt_abs, *
4, 12, division, 16, *, min_max, sqrt_abs

Iteration 160
Generated feature:
11, 21, *, 4, +, 14, mod_column
Prompt examples:
4, log, 13, *, 17, min_max, sqrt_abs, +
6, 12, *, 21, sqrt_abs, +
4, 14, +, 18, *, min_max
11, 21, +, 6, *, sqrt_abs
2, 4, +, 6, division, log
4, 6, +, 16, *, 17, +, min_max
6, 21, division, 17, log, *
12, min_max, 18, *, 5, mod_column
2, 6, mod_column, 17, +, sqrt_abs
2, 3, +, 4, sqrt_abs, *
4, 12, division, 16, *, min_max, sqrt_abs

Iteration 161
Generated feature:
6, 12, +, 17, *, min_max
Prompt examples:

```
6, 12, *, 21, sqrt_abs, +
4, 14, +, 18, *, min_max
11, 21, *, 4, +, 14, mod_column
11, 21, +, 6, *, sqrt_abs
2, 4, +, 6, division, log
4, 6, +, 16, *, 17, +, min_max
6, 21, division, 17, log, *
12, min_max, 18, *, 5, mod_column
2, 6, mod_column, 17, +, sqrt_abs
2, 3, +, 4, sqrt_abs, *
4, 12, division, 16, *, min_max, sqrt_abs
```

### F.4 POTENTIAL FAILURE MODES

One potential failure mode is the generation of features that are duplicates of existing candidate features or syntactically invalid. The third column of each configuration in Table 2 reports the number of LLM generations needed to construct 200 candidate features in our experiments. Specifically, the proportion of valid new features is around 55% using GPT-3.5 and around 60% using GPT-4 on average. Feature search tends to converge per our feature divergence analysis in Section 5.6 and could get stuck in local optima when example features in the prompt are highly similar. From Figure 10, the number of LLM generations needed to construct a new candidate feature remains relatively stable as the algorithm iterates, suggesting low likelihood of getting stuck. Table 3 shows that including more example features in the prompt improves the success rate of feature construction on average by increasing the diversity. Another potential failure mode is that the generated explanation of a constructed feature may be inaccurate, e.g., the column index may be inconsistent with the feature name.

## G   MORE DISCUSSION ON DIFFERENCES FROM EXISTING WORKS

Although our work PromptFE and CAAFE (Hollmann et al., 2023) both utilize LLMs to construct new features incorporating dataset semantic information, they differ in several key aspects. We design PromptFE such that it taps into the in-context learning capability of LLMs and performs effective feature search. In PromptFE, we provide top-performing constructed features in the prompt as learning examples, label them with performance scores, and rank them by score. We demonstrate that the LLM learns to optimize feature construction over the course of algorithm. CAAFE instead stores all previous instructions and code snippets in the conversation history, which hinders the in-context learning of optimal feature patterns. It quickly consumes the LLM's context as the algorithm iterates, incurring more and more LLM generation costs. In comparison, the LLM generation cost of PromptFE stays constant across iterations, without a maximum limit on the number of iterations it can perform. Therefore, our method PromptFE has stronger capability of performing feature search in large search spaces requiring many iterations, such as datasets with numerous feature attributes.

In PromptFE, we also explore representing features in a different form, i.e., canonical RPN (cRPN). We refer to Appendix A for further detail. Compared with the Python code representation in CAAFE, cRPN is more compact, which not only reduces LLM generation costs but also makes the in-context learning of feature patterns easier, and more human interpretable. The use of pre-defined operators reduces the search space and simplifies the learning process for optimizing feature construction. Together, our approach gives better control than code representation that helps avoid undesirable or unexpected LLM outputs. Another advantage of cRPN is that it is convenient to import external features (as outlined in Algorithm 1) and export the results as individual features, providing compatibility with other feature engineering methods.

The rules generated by FeatLLM (Han et al., 2024) are based on a single raw feature, without considering high-order feature interactions. There is no feedback mechanism to improve the rules either. In comparison, our approach constructs new features that combine multiple raw features and iteratively improves feature construction by learning the performance. Furthermore, FeatLLM can be used for only classification tasks and for a single type of downstream model. Our approach is effective for both classification and regression tasks, and it adaptively constructs features that are useful for different downstream models.

ELF-Gym (Zhang et al., 2024b) first generates feature descriptions using one LLM and then generates feature code using another LLM, which is less efficient, and it does not have a feedback mechanism to improve the features. Differently, we represent features in the compact form of cRPNs. Our approach iteratively improves feature construction by learning the performance, where the LLM simultaneously generates new features and explanations on its own.

Compared with OCTree (Nam et al., 2024), our approach represents features concisely in the form of canonical RPNs without using external modules like decision trees. The processes of feature encoding, decoding, and validity check detailed in Algorithms 2 and 3 are simple and efficient. We have demonstrated that cRPNs are effective for the LLMs to understand the structure of features and construct new features to improve utilities. The LLMs are also able to semantically explain the constructed features in the context of dataset descriptive information on its own. Our approach gives better control, such as the number of operators to use to construct features, and facilitates feature search by reducing the search space with a set of pre-defined operators. Moreover, with fewer modules used, our approach is more robust and cost efficient.

While LFG (Zhang et al., 2024a) is also LLM based, it does not utilize the semantic information of datasets. We have shown in Sections 5.3 and 5.5 that the incorporation of dataset semantic information enhances the effectiveness of feature construction of our approach. Informed by dataset semantic information, our approach circumvents exhaustive feature search and evaluates considerably fewer candidate features than traditional approaches, while providing semantic explanations of the constructed features. Another difference is that we represent features in the compact and unambiguous form of cRPNs, which not only reduces LLM generation costs but also facilitates in-context learning of feature patterns. Furthermore, compared with (Zhang et al., 2024a) , we conduct more comprehensive experiments by benchmarking against state-of-the-art LLM- and non-LLM-based AutoFE methods on both regression and classification tasks and perform more detailed performance analysis, such as feature attribute selection, feature complexity, and feature construction efficiency.

More fundamentally, we demonstrate in this work that general-purpose LLMs like GPTs can effectively model recursive tree structures in the form of cRPN feature expressions and reason about the structures in the context of semantic information, shedding light on further LLM-driven applications. We hereby underscore the importance of adopting proper representation for the downstream task to tap into LLMs' potential.

## H  PRACTICAL SIGNIFICANCE

PromptFE constructs semantically meaningful features that significantly boost the performance of simple predictive models especially linear models and provides semantic explanations. Our cRPN feature representation is concise and easy to interpret. Using our approach, one can enhance the performance of simple predictive models without sacrificing their interpretability. Our toolkit is easy to deploy, requiring only OpenAI APIs.

