# OpenReview forum: "PromptFE: Automated Feature Engineering by Prompting"
_ICLR.cc/2026/Conference — ICLR 2026 Conference Withdrawn Submission_

### Official Review · Reviewer_xw8h · 2025-10-25

**Soundness:** 2
**Presentation:** 3
**Contribution:** 2
**Rating:** 4
**Confidence:** 3

**Summary:**

This paper introduces PromptFE, a framework using large language models (LLMs) for automated feature engineering (AutoFE) on tabular data. The method iteratively prompts an LLM (GPT-3.5/4) with dataset/attribute descriptions, a predefined set of operators, and a ranked list of the best-performing features generated so far, represented in canonical Reverse Polish Notation (cRPN). The LLM generates new candidate features in cRPN and provides semantic explanations. Features are evaluated using cross-validation, and the scores guide subsequent LLM prompts via in-context learning. The authors report competitive performance against several AutoFE baselines on public datasets
[Note: I have used LLMs to improve my writing and help me answer paper questions]

**Strengths:**

- Partial Novelty: Uses cRPN to represent features, ensuring uniqueness and providing a structured, concise format potentially easier for the LLM to process and generate compared to code or natural language descriptions.
-Strong Empirical Results: Demonstrates significant performance gains over raw features and achieves competitive or superior average performance against several SOTA AutoFE baselines across multiple datasets and downstream models.

**Weaknesses:**

- Novelty: Many concepts are taken directly from CAAFE with only minor adjustments
- Limited Expressiveness of Feature Space: The chosen feature representation (cRPN) combined with a predefined, relatively basic set of mathematical operators (+, -, *, /, log, sqrt, etc.) significantly restricts the complexity and type of features that can be generated. This approach cannot invent novel, domain-specific transformations (e.g., handling date differences, geospatial calculations, complex conditional logic) that might be crucial for certain datasets. Compared to methods generating arbitrary code, the expressive power of PromptFE's feature space is inherently limited to combinations of these fixed operators, potentially failing to capture more intricate data relationships.
- Scalability Limitations (Cost & Evaluation Overhead): Each iteration involves potentially multiple LLM API calls and, more significantly, feature evaluation using cross-validation. Repeated model training for scoring can be computationally expensive, particularly for complex downstream models or large datasets. The overall wall-clock time might still be substantial.
- Heuristic and Opaque Search: The LLM-driven search is fundamentally heuristic, relying on the model's opaque internal mechanisms to interpret scores and examples. There are no guarantees of convergence, optimality, or systematic exploration. The process might fixate on suboptimal patterns derived from early high-scoring features (local optima). Sensitivity to hyperparameters like temperature and k (number of examples) is also noted.

**Questions:**

Handling Complex Feature Types: How would PromptFE handle the need for features involving date differences, string manipulations, conditional logic (if-then-else), or interactions with external knowledge, given the limitations of the current operator set and cRPN? Is the framework extensible in this regard?

Search Strategy Robustness: How sensitive is the search process to the initial random features, the number of top-k examples (k), and the LLM's temperature? Is there a risk of premature convergence or unproductive exploration?

---

### Official Review · Reviewer_QRWh · 2025-10-31

**Soundness:** 3
**Presentation:** 3
**Contribution:** 3
**Rating:** 6
**Confidence:** 2

**Summary:**

This paper proposes PromptFE, an automated feature engineering framework based on large language models (LLMs). Its core concept involves guiding LLMs to generate new features through prompting, while continuously optimizing feature quality via a context-learning mechanism. PromptFE employs canonical reverse Polish notation (cRPN) as the feature representation format, ensuring uniqueness and parsing capability of feature expressions. Methodologically, PromptFE integrates dataset descriptions, field semantics, operator definitions, and high-scoring feature examples into prompts. It iteratively generates and evaluates new features, ultimately selecting the optimal feature subset through cross-validation. Experiments across seven real-world datasets compare PromptFE against four baseline methods including traditional AutoFE and LLM-based approaches, encompassing both linear and nonlinear models. Results validate PromptFE's advantages in performance, efficiency, and interpretability.

**Strengths:**

(1) Method design is simple yet effective: PromptFE ingeniously combines the context learning capability of LLMs with feature engineering tasks. By representing features through cRPN, it avoids expression ambiguity and enhances the model's understanding of feature structures and generation quality.
(2) Comprehensive experimentation with thorough comparisons: Systematic evaluations against mainstream AutoFE methods like OpenFE, DIFER, and CAAFE across multiple public datasets—encompassing diverse model types—consistently demonstrate PromptFE's significant performance advantages, achieving over 15% improvement specifically on linear models.
(3) High interpretability with semantic utilization: PromptFE not only generates features but also provides semantic explanations, enhancing feature comprehensibility and credibility. Additionally, ablation experiments validate the critical role of dataset semantic information in feature quality.

**Weaknesses:**

(1) Computational efficiency still has room for optimization: Although PromptFE evaluates far fewer features than traditional methods, each round still requires invoking the LLM and training the model for evaluation, resulting in relatively high overall computational costs. We recommend introducing early-stopping mechanisms or uncertainty-based feature selection strategies to reduce redundant evaluations.
(2) Strong dependency on LLMs: The current approach relies heavily on the generative capabilities and semantic understanding of GPT-3.5/GPT-4, without fully exploring the applicability of smaller or locally deployed models. This may limit its use in resource-constrained scenarios. Testing lightweight models like LLaMA-7B is recommended to enhance the method's versatility and controllability.
(3) Lack of stability analysis for generated features: While experiments demonstrate significant performance gains, systematic analysis of feature stability across different random seeds or data partitions is absent. We recommend incorporating feature consistency evaluations (e.g., Jaccard similarity, feature importance stability) to strengthen robustness validation.

**Questions:**

(1) Does PromptFE maintain its current efficiency and performance advantages on larger datasets (e.g., millions of samples)? Are there corresponding complexity analyses or scalability experiments?
(2) The current method primarily relies on GPT-series models. Have performance differences been considered when using open-source LLMs like LLaMA or Qwen with identical prompts? Are there impacts from model bias or variations in semantic understanding?
(3) Does PromptFE support user-defined operators or domain-specific knowledge embedding? Are there future plans to introduce a more flexible, extensible framework to accommodate domain-specific requirements?

---

### Official Review · Reviewer_hJja · 2025-11-04

**Soundness:** 2
**Presentation:** 2
**Contribution:** 2
**Rating:** 2
**Confidence:** 5

**Summary:**

This paper proposes an automatic feature engineering framework PromptFE. This work uses dataset metadata and feature correlations to generate new features. The core is to adopt LLM reasoning to incorporate dataset descriptions into prompts to select features. It further introduces a compact feature representation to iteratively produce feature-enhancement code based on a predefined set of operators. Experimental results show that this method improves feature quality and model performance compared to baseline approaches.

**Strengths:**

The motivation of this paper is sound. Automated feature engineering should be more explainable and visible, and using LLMs to generate features is an interesting direction.

The workflow is well-structured, as it defines a compact feature representation and integrates predefined transformation operators to make the feature generation process interpretable and reproducible.

The experimental evaluation includes comparisons with some existing methods, the proposed method achieves better performance in the given benchmark cases. The paper also provides illustrative examples.

**Weaknesses:**

The novelty is limited. Although explained (in the appendix), I still do not see much difference between this work and CAFFE, or the LFG.

The algorithm does not match well with the method and problem definition, and is not clearly illustrated individually.

For experiments, I do not see much performance gains. The experiments were on GPT3.5 and GPT4 only, more models, especially white box models, like Llama or Qwen, should be tested. Also there should be an ablation study to prove the effectiveness of each component.

**Questions:**

See above. And the figures in Figure 6 are really small. I use my largest screen, still find them hard to read.

---

### Official Review · Reviewer_X4Lj · 2025-11-09

**Soundness:** 2
**Presentation:** 2
**Contribution:** 2
**Rating:** 2
**Confidence:** 4

**Summary:**

This paper proposes an automated feature engineering method using large language models (LLMs). The proposed method, termed PromptFE, automatically constructs features in a compact string format by using reverse Polish notation and generates semantic explanations based on dataset descriptions. The experimental result demonstrates that PromptFE outperforms existing automated feature engineering methods on tabular datasets.

**Strengths:**

- The proposed method leverages canonical Reverse Polish Notation (cRPN) to represent features.
- The effectiveness of the proposed method is verified through experiments on several tabular datasets.

**Weaknesses:**

- Overall framework of the proposed method is similar to existing methods such as CAAFE (Hollmann et al., 2023). The primary difference lies in the use of RPN for feature representation. The novelty of the proposed method seems to be limited.
- It is not clear whether RPN is really effective for feature representation in the LLM-based AutoFE. It would be better if the authors could compare RPN with other representations, such as mathematical expressions or Python code, under the same prompt template setting.
- The number of datasets used in the experimental evaluation seems small compared to prior works.

**Questions:**

- The reviewer wonders about the description "PromptFE reduces the search space with pre-defined operators and represents features in compact cRPN." How does the use of cRPN reduce the search space? What is the definition of the search space in this context? Is it possible to quantify the reduction of the search space by using cRPN?
- How is the performance of the proposed PromptFE using mathematical expressions or Python code representation instead of cRPN? This comparison will help in understanding the effectiveness of cRPN for feature representation.
- Could you clarify the selection reason for the datasets used in the experimental evaluation?
- The title of this paper seems ambiguous. It might be better to include specific words that reflect the key contribution.

---

### Note · Authors · 2025-12-15

I have read and agree with the venue's withdrawal policy on behalf of myself and my co-authors.